

# Exploring the potential of the nano-Köhler theory to describe the growth of atmospheric molecular clusters by organic vapors

Jenni Kontkanen[1,2], Tinja Olenius[1], Markku Kulmala[2,3,4], and Ilona Riipinen[1]

[1]Department of Environmental Science and Analytical Chemistry (ACES) and Bolin Centre for Climate Research, Stockholm University, Stockholm, Sweden
[2]Institute for Atmospheric and Earth System Research / Physics, Faculty of Science, University of Helsinki, Helsinki, Finland
[3]Aerosol and Haze Laboratory, Beijing Advanced Innovation Center for Soft Matter Science and Engineering, Beijing University of Chemical Technology, Beijing, P.R. China
[4]Joint International Research Laboratory of Atmospheric and Earth System Sciences, School of Atmospheric Sciences, Nanjing University, Nanjing, P.R. China

*Correspondence to*: Jenni Kontkanen (jenni.kontkanen@helsinki.fi)

**Abstract.** New particle formation involving sulfuric acid, bases and oxidized organic compounds is an important source of atmospheric aerosol particles. One of the mechanisms suggested to depict this process is the nano-Köhler theory, which describes the activation of inorganic molecular clusters to growth by a soluble organic vapor. In this work we studied the capability of the nano-Köhler theory to describe the growth of atmospheric molecular clusters by simulating the dynamics of a cluster population in the presence of a sulfuric acid–base mixture and an organic compound. We observed nano-Köhler type activation in our simulations when the saturation ratio of the organic vapor and the ratio between organic and inorganic vapor concentrations were in a suitable range. However, the nano-Köhler theory was unable to predict the exact size at which the activation occurred in the simulations. In some conditions apparent cluster growth rate (GR) started to increase close to the activation size determined from the simulations. Nevertheless, because the behavior of GR is also affected by other dynamic processes, GR alone cannot be used to deduce the cluster growth mechanism.

## 1 Introduction

Atmospheric new particle formation (NPF) is a significant source of global cloud condensation nuclei (CCN) (Kerminen et al., 2012; Spracklen et al., 2008), and thus it affects the magnitude of indirect aerosol forcing (Kazil et al., 2010; Makkonen et al., 2012). NPF has also been observed to contribute to particulate pollution events in urban environments, such as Chinese megacities (Guo et al., 2014). Therefore, accurate understanding of NPF, including the formation of the initial nanometer-sized molecular clusters and their further growth to larger particles, is necessary both to decrease uncertainties in climate projections, and to tackle urban air quality problems.



According to current knowledge, sulfuric acid is a major driver of the first steps of NPF in most environments (Kuang et al., 2008; Paasonen et al., 2010; Kulmala et al., 2013). Base compounds, such as ammonia and amines, are also important due to their ability to cluster efficiently with sulfuric acid and stabilize the initial nanoparticles (Almeida et al., 2013; Jen et al., 2014, 2016; Kirkby et al., 2011). Low-volatile organic vapors have long been considered as responsible for growing the freshly-

formed particles to larger sizes (e.g. Kuang et al., 2012; Riipinen et al., 2012; Yli-Juuti et al., 2011). Furthermore, recent experimental results show that highly oxidized organic species, formed by oxidation of monoterpenes, can participate in NPF already at nanometer sizes (Schobesberger et al., 2013; Kirkby et al., 2016). However, understanding of the physics and chemistry of NPF involving organic compounds is still very limited, largely because the identities and properties of the organic species are poorly known (Donahue et al., 2013; Elm et al., 2017). In particular, there is a lack of a robust physical description

of the formation of the initial nanoparticles in the presence of sulfuric acid, bases and organic compounds. Such a description is needed for correct interpretation of the experimental data on NPF, but also for implementing the process accurately in larger-scale atmospheric transport models.

NPF is often described as a two-step process (Kulmala et al., 2000). In the first step, a stable cluster that is more likely to grow than to evaporate is formed by nucleation, and in the second step the cluster grows to larger sizes by condensation. The latter

step is often referred to as activation of the clusters. Accordingly, particle formation involving sulfuric acid, bases, and low-volatile organic compounds can be considered to proceed via the formation of a stable sulfuric acid-base cluster, followed by its growth by condensation of organic vapor (Kulmala, 2004). However, if the participation of organic vapor in NPF occurs simply by condensation, the organic vapor needs to have an extremely high saturation ratio to be able to contribute to the growth at nanometer sizes (Kulmala et al., 2004). This is due to the large surface-to-volume ratio of the clusters, which makes

attachment of molecules thermodynamically unfavorable. This is qualitatively depicted by the Kelvin effect, in which the saturation ratio over the surface of a spherical particle increases with decreasing particle size. Several mechanisms have been proposed to describe how nanometer-sized clusters can overcome this thermodynamic barrier. These mechanisms include heterogeneous nucleation of organic vapors (Wang et al., 2013), heterogeneous reactions between clusters and organic vapors (Wang et al., 2010; Zhang and Wexler, 2002), adsorption of organics on cluster surface (Wang and Wexler, 2013) and a nano-

Köhler-type mechanism (Kulmala et al., 2004). In this study, we focus on the nano-Köhler mechanism and its capability to describe the growth of atmospheric clusters.

The nano-Köhler theory was first proposed by Kulmala et al. (2004). It describes the activation of nanometer-sized inorganic clusters to growth by condensation of organic vapor, which is soluble in the inorganic compound and water. Thus, the nano-Köhler theory is analogous to the Köhler theory describing the activation of CCN to cloud droplets. The fact that the nano-

Köhler theory is based on the macroscopic properties of the condensing vapors makes it appealing to use for describing the initial particle formation. The nano-Köhler mechanism has been applied in aerosol dynamics models (Anttila et al., 2004; Korhonen et al., 2004), used to describe activation of particles inside condensation particle counters (Giechaskiel et al., 2011; Kulmala et al., 2007), and suggested to explain observed behavior of sub-5 nm particle populations in field and laboratory measurements (Kuang et al., 2012; Kulmala et al., 2013; Tröstl et al., 2016). In a recent laboratory study Tröstl et al. (2016)



examined particle growth rates (GR) in the presence of monoterpene oxidation products and observed accelerating growth at sizes below 5 nm. Using a volatility distribution growth model, where the condensational growth of a single representative particle is described by the nano-Köhler mechanism, the authors suggest that the particle growth at sizes below ~2 nm is governed by organic vapors with extremely low volatility. At larger sizes, where the Kelvin effect is reduced, more abundant vapors with slightly higher volatilities can start to contribute to the growth, resulting in a higher GR.

Although the nano-Köhler theory has been applied in aerosol dynamics and single-particle growth models, these modelling approaches provide only limited knowledge of the capability of the theory to describe the very first steps of NPF. This is because these models describe condensation macroscopically, which means that all particles of a given size grow or shrink at the same rate. However, the formation and growth of the smallest sizes is affected by stochastic collision and evaporation processes, which causes widening of the nanoparticle size distribution and enables the crossing of thermodynamic barriers by homogeneous nucleation (Wang et al., 2013). Furthermore, single-particle modeling approaches assume that particles grow only by vapor monomer collisions, while both experimental and modeling studies indicate that cluster-cluster collisions may significantly contribute to particle growth in some conditions (Kontkanen et al., 2016; Lehtipalo et al., 2016). Modeling the initial NPF accurately requires using a cluster kinetics model, which simulates the time-development of cluster concentrations explicitly considering stochastic effects and including all collision and evaporation processes between vapor molecules and clusters (Olenius and Riipinen, 2017).

In this study, our aim is to investigate the potential of the nano-Köhler theory to describe the growth of atmospheric molecular clusters by organic vapors. For this we use a molecular-resolution model, which allows us to explicitly simulate the time-evolution of a cluster population involving organic and inorganic species. First, we discuss similarities and differences between the nano-Köhler theory and the traditional Köhler theory and compare their assumptions to real atmospheric molecular systems. Then, we apply cluster kinetics simulations to study in what kind of conditions nano-Köhler type behavior can be observed. Specifically, we investigate the effects of vapor properties, such as volatility and vapor concentrations, on the dynamics of the cluster population. We also compare the results on cluster activation obtained from the simulations to the predictions of the nano-Köhler theory, and assess to what extent this simplified theory is able to capture the cluster growth mechanisms.

## 2 Theory

The nano-Köhler theory describes the activation of inorganic clusters to growth by spontaneous condensation of organic vapor, which is soluble in the inorganic compound and water (Kulmala et al., 2004). In the original nano-Köhler theory, presented by Kulmala et al. (2004), a system involving sulfuric acid–ammonia clusters, an organic compound, and water is studied. However, the theory can also be applied to other seed compositions and to systems containing only inorganic clusters and an organic vapor without water, which is the case in this study.





The nano-Köhler theory is based on assuming a thermodynamic equilibrium between the clusters and the condensing organic vapor. In an analogous way to the traditional Köhler theory, the equilibrium saturation ratio of organic compound ($S_{\mathrm{org,eq}}$) is obtained from

$$S_{\mathrm{org,eq}} = x_{\mathrm{org}} a_{\mathrm{org}} \exp\left(\frac{4\sigma_{\mathrm{org}} m_{\mathrm{org}}}{k_{\mathrm{B}} T \rho_{\mathrm{org}} d_{\mathrm{p}}}\right), \qquad\qquad (1)$$

where $x_{\mathrm{org}}$ is the molar fraction of the organic compound in the particle phase, $a_{\mathrm{org}}$ is the activity coefficient, $m_{\mathrm{org}}$ is the molecule mass, $\rho_{\mathrm{org}}$ is the liquid-phase density, $\sigma_{\mathrm{org}}$ is the surface tension, and $d_{\mathrm{p}}$ is the cluster diameter. $k_B$ is the Boltzmann constant and $T$ is temperature. A similar equation can be written for the equilibrium saturation ratio of water if it is included in the studied system (Kulmala et al., 2004).

From Eq. (1) it is possible to solve the equilibrium saturation ratio of organic compound for different cluster sizes $d_{\mathrm{p}}$ in the
presence of a seed cluster with diameter $d_{\mathrm{p,s}}$. When the resulting equilibrium saturation ratios are plotted as a function of the cluster size, a curve similar to traditional Köhler curves is obtained. The curve exhibits a peak resulting from the combination of Kelvin and Raoult effects, caused by the curvature of the cluster surface and solubility of the compounds (Vehkamäki and Riipinen, 2012). When the saturation ratio of the organic compound is lower than the peak value, the seed clusters trapped behind the peak maintain their equilibrium size but they cannot grow further. When the saturation ratio increases beyond the
peak value, the clusters start to grow spontaneously by condensation of organic vapor (see Fig. 1). Thus, the ascending part of the curve corresponds to the stable equilibrium state and the descending part to the unstable equilibrium. Note that this behavior is different than in heterogeneous nucleation, where the seed is insoluble in the nucleating vapor, and the saturation ratio needed for the cluster activation decreases as a function of cluster size without a maximum (Winkler et al., 2008).

Although the thermodynamic equilibrium is described in an analogous way in the nano-Köhler theory and in the traditional
Köhler theory, there are differences between these two theories, especially related to the systems that they are describing (Anttila et al., 2004; Kulmala, 2004). The comparison between the theories is presented in Table 1. Furthermore, the real systems differ significantly from the simplified system assumed in the nano-Köhler theory (see Table 1 and Fig. 1). In real systems there is no specific non-evaporating seed or condensing vapor but there is a distribution of inorganic and organic vapor molecules and clusters which all can collide and evaporate. Furthermore, there does not necessarily exist an energy barrier that
needs to be overcome, but the growth of clusters may be entirely governed by kinetic collisions. If an energy barrier exists, clusters may overcome the barrier even before the critical saturation ratio is reached as a result of stochastic collisions, i.e. nucleation, and thus the system is not in thermodynamic equilibrium all the time (Vehkamäki and Riipinen, 2012). In addition, in real systems organic and inorganic vapors are not lost solely due to scavenging by pre-existing particle population but also due to collisions with clusters. Furthermore, vapor concentrations and external losses may vary over time, which can affect the
dynamics of the cluster population.



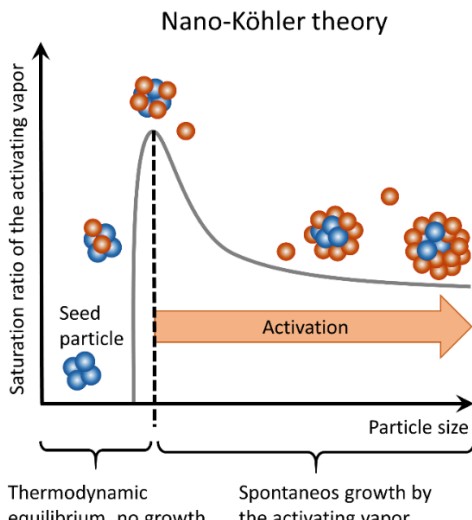
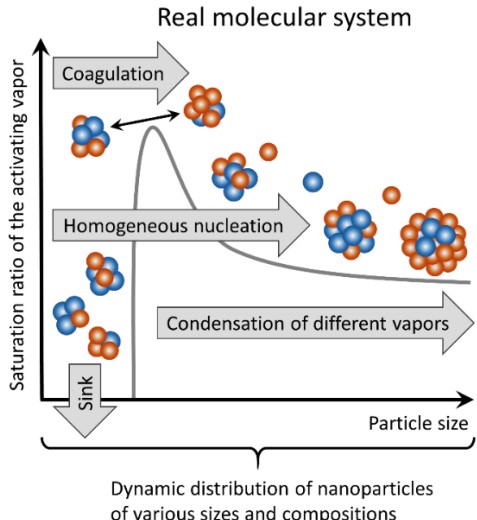

**Figure 1: Schematic figure illustrating how the growth of atmospheric molecular clusters is described by the nano-Köhler theory, and how the growth occurs in real molecular systems affected by various dynamic processes.**

**Table 1: Comparison between the Köhler theory, the nano-Köhler theory and real systems of atmospheric molecular clusters.**

| *Köhler theory* | *Nano-Köhler theory* | *Real molecular systems* |
|---|---|---|
| Describes the activation of CCN ($d_P$>50 nm) to cloud droplets by spontaneous condensation of water vapor. | Describes the activation of inorganic clusters ($d_p$ = ~1–3 nm) for growth by spontaneous condensation of organic vapor. | A distribution of clusters of varying sizes and compositions including inorganic and organic compounds can exist simultaneously at all times. |
| The condensing vapor is water vapor with typical atmospheric concentrations of ~$10^{17}$ cm$^{-3}$. | The condensing vapor is a water-soluble organic compound with concentrations likely ranging from ~$10^5$ to $10^8$ cm$^{-3}$ (Jokinen et al., 2017). | Vapor concentrations are not constant but may vary over time. |
| The seed consists of a mixture of inorganic/organic compounds and is water-soluble. The seed compounds do not evaporate. | The seed consists of sulfuric acid and bases and is soluble in the condensing organic compound. The seed compounds do not evaporate but condense irreversibly on the cluster. | There is no seed in the same sense as in the theory. Both inorganic and organic compounds can condense and evaporate and may contribute to the growth. |
| Thermodynamic equilibrium between water and the seed particle is assumed. | Thermodynamic equilibrium between the organic compound, the seed cluster and water is assumed. The energy barrier width in the nano-Köhler is very narrow with respect to the number of molecules compared to the Köhler theory, and thus addition of only few molecules may result in overcoming the barrier. | Clusters can nucleate over barriers and they may not be in thermodynamic equilibrium before activation to growth. |
| The growing cloud droplets scavenge the available water vapor thereby limiting the activation process. | The loss rate of organic vapor is determined mainly by larger background aerosol particles, and not the growing clusters. | The cluster population is affected by losses due to background particles, and cluster self-coagulation may also be important. The magnitude of external losses may vary over time. |



## 3 Methods

### 3.1 Cluster kinetics model

We used a molecular-resolution cluster kinetics model to simulate the time-development of atmospheric cluster concentrations. Two model compounds, one corresponding to an inorganic compound and one corresponding to an organic compound, were used in the simulations (see the next section for the properties of the model compounds). The cluster population was simulated starting from vapor monomers up to clusters with mass diameter of 3.2–3.4 nm, including all possible cluster compositions. The largest simulated clusters correspond to the size of a pure inorganic cluster composed of 100 molecules.

In the model, the discrete General Dynamics Equation (GDE; Friedlander, 1977) is numerically solved for each cluster composition $i$ including all processes where a cluster can be formed or lost:

$$\frac{dC_i}{dt} = \frac{1}{2}\sum_{j<i}\beta_{j,(i-j)}C_jC_{i-j} + \sum_j \gamma_{(i+j)\to i,j}C_{i+j} - \sum_j \beta_{i,j}C_iC_j - \frac{1}{2}\sum_{j<i}\gamma_{i\to j,(i-j)}C_i + Q_i - L_iC_i \ . \tag{2}$$

Here $C_i$ is the concentration of cluster $i$ and $\beta_{i,j}$ is the collision rate coefficient between cluster $i$ and cluster $j$. $\gamma_{(i+j)\to i,j}$ is the evaporation rate coefficient of cluster $(i+j)$ to clusters $i$ and $j$, which we considered only for evaporation of vapor monomers. $Q_i$ is the source rate, here included only for vapor molecules. $L_i$ is the loss rate coefficient corresponding to the external sink for vapors and clusters. The set of GDEs was generated and solved with the ACDC (Atmospheric Cluster Dynamics Code) program (Olenius and Riipinen, 2017).

The collision rate coefficients $\beta_{i,j}$ were calculated assuming hard-sphere collisions. The evaporation rate coefficients of vapor monomers were calculated according to the Kelvin formula:

$$\gamma_{(i+j)\to i,j} = \beta_{i,j}\frac{p_{sat,i}}{k_B T}x_i \exp\left(\frac{4\sigma m_i}{k_B T \rho_i d_{pj}}\right) \tag{3}$$

Here $\beta_{i,j}$ is the collision rate coefficient between vapor compound $i$ and cluster $j$. $p_{sat,i}$ is the saturation vapor pressure of the compound, $x_i$ is the molar fraction of the compound in cluster $j$, $\sigma$ is the cluster surface tension, $\rho_i$ is the liquid phase density of $i$, and $d_{p,j}$ is the cluster diameter. $k_B$ is the Boltzmann constant and $T$ is the temperature. Calculating evaporation rates from Eq. (3) is consistent with calculating equilibrium saturation ratios from Eq. (1) when assuming that the activity coefficient $a = 1$.

In most simulations the external loss coefficient $L_i$ was set to correspond to typical losses in a chamber experiment, including a size-dependent wall loss ($L_{\text{wall}}$) and a size-independent dilution loss ($L_{\text{dil}}$) according to Kürten et al. (2015):

$$L_i = L_{\text{wall}} + L_{\text{dil}} = \frac{A_{wall}}{d_{p,i}} + L_{\text{dil}} \tag{4}$$

Here the empirical constant $A_{wall}$ is 0.001 nm s⁻¹ and the loss rate due to dilution $L_{\text{dil}}$ is $9.6 \cdot 10^{-5}$ s⁻¹ (Kürten et al., 2015).

In one simulation set, external losses corresponding to a sink caused by a background particle population in the planetary boundary layer were used. In this case, the loss coefficient was obtained from (Lehtinen et al., 2007):

$$L_i = L_{ref} \cdot \left(\frac{d_p}{d_{p,ref}}\right)^b \tag{5}$$



Here $L_{ref}$ is the loss coefficient for a reference cluster, which in our case is the inorganic vapor monomer, and $d_{p,ref}$ is the diameter of the reference cluster. We set the reference loss coefficient to $10^{-3}$ s$^{-1}$ and the exponent $b$ to $-1.6$, which represent typical values in a boreal forest (Lehtinen et al., 2007).

## 3.2 Studied compounds

We simulated binary mixtures of inorganic and organic compounds relevant for atmospheric particle formation. The inorganic compound corresponds to a quasi-unary sulfuric acid–base mixture, and the organic compound represents atmospheric oxidized organic species (see Table 2). All model compounds were assumed to consist of spherical molecules.

In most simulations the inorganic compound was set to have the mass of a sulfuric acid–dimethylamine cluster (SA–DMA) and $p_{sat} = 0$. Additional simulations were performed using the mass of ammonium bisulfate (SA–NH$_3$) and $p_{sat} = 10^{-9}$ Pa. The

properties of these model substances are consistent with observations on a 1:1 molar ratio in which dimethylamine and ammonia cluster with sulfuric acid, dimethylamine forming seemingly non-evaporating clusters (Jen et al., 2014; Kürten et al., 2014).

In most simulations, the organic compound (ORG) was set to have a mass of 300 amu and its $p_{sat}$ was varied between $10^{-4}$ and $10^{-12}$ Pa. Of these saturation vapor pressures, two were selected for a more detailed study: $p_{sat} = 10^{-8}$ Pa, representing a low-

volatile organic compound (LVOC), and $p_{sat} = 10^{-11}$ Pa, representing an extremely low-volatile organic compound (ELVOC). In addition, simulations were performed using a compound with $p_{sat} = 10^{-8}$ Pa and a mass of 500 amu, corresponding to LVOC with a larger mass (LVOC$_{large}$). The saturation vapor pressures of the LVOC and ELVOC species are consistent with the volatility basis set classification of organic compounds with low and extremely low volatilities (Donahue et al., 2012). They also may represent typical saturation vapor pressures of alpha-pinene oxidation products (Tröstl et al., 2016) but as different

methods give very different estimates for saturation vapor pressures, the exact values are uncertain (Kurtén et al., 2016).

In addition to the saturation vapor pressures and molecular masses, the assumed liquid phase densities, the particle surface tension and the activity coefficients can have significant effects on the results. In earlier studies investigating the nano-Köhler theory, these properties have been obtained by fitting a simple thermodynamic model to measured data on particle properties (Anttila et al., 2004; Kulmala, 2004). However, because of large uncertainties related to estimating these properties for

nanometer-sized clusters, we chose to use a simple approach, and set all particles to have a density of 1500 kg m$^{-3}$ and a surface tension of $2.3 \cdot 10^{-2}$ N m$^{-1}$. Activity coefficients were set to 1, corresponding to an ideal solution. Furthermore, we do not include water in the model system although it is incorporated in the original nano-Köhler theory. Including water would lower the saturation vapor pressure of the organic compound through the solute effect (Raoult's law). We address the effect of the organic compound volatility by performing simulations with a large range of organic saturation vapor pressures.




**Table 2: Model compounds, their molecular mass ($m$), saturation vapor pressure ($p_{sat}$) and the corresponding number concentration ($C_{sat}$) and mass concentration ($C_{mass,sat}$) at $T = 278$ K. Note that LVOC and ELVOC are two cases of ORG selected for more detailed studies.**

| Model substance | $m$ (amu) | $p_{sat}$ (Pa) | $C_{sat}$ (cm$^{-3}$) | $C_{mass,sat}$ (µg m$^{-3}$) |
|---|---|---|---|---|
| SA–DMA | 143 | 0 | 0 | 0 |
| SA–NH$_3$ | 115 | $10^{-9}$ | $2.6 \cdot 10^5$ | $5.0 \cdot 10^{-5}$ |
| ORG | 300 | $10^{-12}$–$10^{-4}$ | $2.6 \cdot 10^2$–$2.6 \cdot 10^{10}$ | $1.3 \cdot 10^{-7}$–$1.3 \cdot 10^1$ |
| LVOC | 300 | $10^{-8}$ | $2.6 \cdot 10^6$ | $1.3 \cdot 10^{-3}$ |
| ELVOC | 300 | $10^{-11}$ | $2.6 \cdot 10^3$ | $1.3 \cdot 10^{-6}$ |
| LVOC$_{large}$ | 600 | $10^{-8}$ | $2.6 \cdot 10^6$ | $2.6 \cdot 10^{-3}$ |

### 3.3 Simulation sets

All performed simulation sets are described in Table 3. In simulation sets 1–4, the concentrations of the inorganic vapor ($C_{SA}$) and the organic vapor ($C_{ORG}$) were set to constant values and the simulations were run until the steady state was reached. In addition, in order to mimic a laboratory set-up, simulations with constant vapor source rates, resulting in the same steady-state vapor concentrations, were performed to study the cluster growth rates (GRs). By default, the inorganic and organic species were SA–DMA and LVOC or ELVOC, unless otherwise stated. Temperature was set to 278 K in all the simulations. The sets

address the effect of the volatility $p_{sat,ORG}$ and concentration $C_{ORG}$ of the organic vapor (set 1), the effect of relatively high sulfuric acid concentration $C_{SA}$ (set 2), the effect of the properties of the inorganic vapor (set 3), and the effect of the mass of the organic vapor (set 4). The simulation set 5 addresses the effect of time-dependent vapor concentration profiles corresponding to atmospheric conditions. Two different simulations were performed: 1) a simulation where inorganic and organic vapor source rates exhibit maxima, 2) a simulation where the inorganic vapor source rate exhibits a maximum but the

organic vapor source rate is constant. These simulations were run for 12 hours.



**Table 3: Description of the simulation sets. The variables are varied at intervals of one order of magnitude. Note that the presented vapor concentrations are steady-state values for sets 1–4 and the maximum values for set 5.**

| Simulation set | Compounds | Vapor concentrations (cm$^{-3}$) | Conditions |
|---|---|---|---|
| 1 | SA–DMA | $C_{SA} = 10^6$ | constant $C_{SA}$ and $C_{ORG}$, |
| | ORG, $p_{sat} = 10^{-12}\ldots10^{-6}$ | $C_{ORG} = 10^4\ldots10^{10}$ | chamber experiment losses (Eq. 4) |
| 2 | SA–DMA | $C_{SA} = 10^7$ | constant $C_{SA}$ and $C_{ORG}$, |
| | LVOC or ELVOC | $C_{ORG} = 10^7\ldots10^{10}$ | chamber experiment losses (Eq. 4) |
| 3 | SA–NH3 | $C_{SA} = 10^6$ | constant $C_{SA}$ and $C_{ORG}$, |
| | LVOC | $C_{ORG} = 10^6\ldots10^9$ | chamber experiment losses (Eq. 4) |
| 4 | SA–DMA | $C_{SA} = 10^6$ | constant $C_{SA}$ and $C_{ORG}$, |
| | LVOC$_{large}$ | $C_{ORG} = 10^6\ldots10^9$ | chamber experiment losses (Eq. 4) |
| 5 | SA–DMA | $C_{SA,\ max} = 10^6$ | varying $C_{SA}$ and $C_{ORG}$, |
| | LVOC | $C_{ORG,\ max} = 5\cdot10^6,\ 10^7$ | background particle losses (Eq. 5) |

### 3.4 Analysis of simulated data

From the simulated cluster concentrations, we determined the contributions of different vapor monomers and clusters to the growth over selected threshold sizes $d_p^{th}$ between 1 and 3 nm. The net cluster flux $J_{d_p^{th}}$ past each threshold size was determined by considering all the collisions and evaporations between different clusters or vapor molecules that lead to crossing of the threshold as

$$J_{d_p^{th}} = \frac{1}{2}\sum(\beta_{i,j}C_iC_j - \gamma_{i+j\to i,j}C_{i+j}),\qquad(6)$$

where $d_{p,i}$ and $d_{p,j} < d_p^{th}$, and $d_{p,i+j} \geq d_p^{th}$. The contribution of an individual molecule or cluster to this flux was determined so that for each collision-evaporation process between species $i$ and $j$ leading to the crossing of the threshold, the resulting flux was attributed to the species with the smaller number of molecules. The reasoning behind this is that when, for example, a vapor monomer $i$ collides with cluster $j$ composed of several molecules, the monomer is causing the cluster to grow in size,

and thus the flux is assigned to be due to $i$.

In addition, we determined the apparent cluster growth rates (GR) using the experimental approach that is often applied for measured particle size distribution data (Lehtipalo et al., 2014). In order to treat the simulation data similarly to measured particle concentrations, we divided the clusters into linearly-spaced size bins of a width of 0.1 nm based on their mass diameter. Vapor monomers were, however, omitted from the size bins to ensure that they do not dominate the smallest bins. Then, we

determined the appearance time $t_{app}$ for each size bin as the time at which the concentration of the bin reaches 50% of the total



concentration increase in the bin (Lehtipalo et al., 2014). Finally, we determined GR for each bin $k$ with the mean diameter $d_{p,k}$ by numerically differentiating the ($t_{app}$, $d_p$)-data:

$$GR_k = \frac{d_{p,k+1} - d_{p,k}}{t_{app,k+1} - t_{app,k}} \tag{7}$$

**3.5 Nano-Köhler calculations**

To compare the predictions of the nano-Köhler theory to the results of the cluster kinetics simulations, we calculated the equilibrium saturation ratio of the organic vapor from Eq. (1) and determined the activation diameter and the corresponding saturation ratio for different seed cluster sizes. In these calculations, we used the same properties of the compounds as in the cluster population simulations (see Sect. 3.2), and varied the seed cluster size between 0.9 and 2.9 nm.

**4. Results and discussion**

The cluster kinetics simulations indicate that nano-Köhler-type activation occurs in specific conditions, determined by vapor concentrations and the organic vapor saturation ratio. In this section, we first go through two example cases: one case where nano-Köhler-type activation is observed, and another case without activation. Then we discuss more generally the conditions where activation occurs, and compare the activation sizes determined from the simulations to those based on the nano-Köhler theory. We also study the connection between the activation sizes and the behavior of apparent cluster GRs. Finally, we discuss

the sensitivity of the results to vapor properties and the time-evolution of the vapor concentrations.

**4.1 Effects of organic vapor volatility and vapor concentrations on the growth mechanism**

**4.1.1 Simulations with LVOC**

Figures 2–4 show the results for the simulations where the model compounds were SA-DMA and LVOC, $C_{SA} = 10^6$ cm$^{-3}$ and $C_{ORG} = 10^6$–$10^9$ cm$^{-3}$. Figure 2 presents the steady-state cluster distributions, illustrating how the relative contributions of SA

and LVOC to the growth of the cluster population depend on their concentrations. When both $C_{ORG}$ and $C_{SA}$ are $10^6$ cm$^{-3}$, the growth proceeds mainly by additions of SA molecules, while at $C_{ORG} = 10^7$ cm$^{-3}$, both SA and LVOC participate in the growth. At $C_{ORG} = 10^8$ cm$^{-3}$ or higher, the growth is clearly dominated by LVOC.

A more detailed picture of the roles of SA and LVOC in the cluster growth is obtained by studying the particle fluxes past selected threshold sizes caused by different vapor monomers or clusters. Figure 3 shows the steady-state flux fractions for five

threshold sizes between 1 and 3 nm in different simulations. The flux fractions are presented for SA monomer, SA dimer, pure SA clusters (trimers or larger), LVOC monomer, LVOC dimer, pure LVOC clusters (trimers or larger), and for the clusters containing both SA and LVOC (dimers or larger).



When $C_{ORG} = 10^6$ cm$^{-3}$, SA monomer dominates the flux at all sizes, and the rest of the flux is mainly caused by SA dimer and SA clusters. The contribution of LVOC monomer to the flux is minor at all sizes. When $C_{ORG} = 10^7$ cm$^{-3}$, the fractions of the flux due to SA and LVOC exhibit a distinct size-dependency. The contribution of SA monomer to the flux is highest at 1.5 nm and lowest at 3 nm. LVOC has a significant contribution to the flux at 1 nm, a minor contribution at 1.5 nm, and a larger,

increasing contribution at 2 nm and above. This behavior resembles the nano-Köhler type activation where the critical size, corresponding to the peak of the Köhler curve, is around 2 nm; at sizes smaller than that, the growth by organic vapor is minor due to the Kelvin barrier, while at larger sizes, organic vapor can start to spontaneously condense on the clusters. When interpreting the cluster flux results it should be kept in mind, though, only the collisions between vapor monomers and the smallest clusters can contribute to the flux past 1 nm due to their small size.

When $C_{ORG} = 10^8$ cm$^{-3}$, LVOC monomer dominates the flux at 1 nm as well as at 2 nm and above. However, at 1.5 nm, most of the flux is caused by LVOC clusters, and the net flux of LVOC monomer is negative due to its high evaporation flux. Thus, this can also be considered as nano-Köhler type situation where the critical size is around 2 nm, and at sizes just below that the growth can occur only by self-coagulation, and not by condensation of vapor monomers. It should be noted, though, that large uncertainties are related to calculating the evaporation rate of LVOC monomers from Kelvin formula (Eq. 3). Therefore,

LVOC dimers and other small clusters may become unrealistically stable compared to monomers, and their contribution to the growth may be overestimated. Evaporation rates assessed by more sophisticated methods, namely quantum-mechanics-based approaches, also exhibit very high quantitative uncertainties (see e.g. Elm et al., 2017 and references therein).

When $C_{ORG} = 10^9$ cm$^{-3}$, LVOC monomer contributes to the flux at all sizes. The fluxes due to LVOC dimer and LVOC clusters are also significant at sizes above 1 nm. Thus, this simulation corresponds to the situation where the Kelvin barrier does not

exist at any size, due to high vapor concentration.

We also studied GRs determined for different size bins based on their appearance times (Fig. 4). When $C_{ORG} = 10^6$ cm$^{-3}$, GR varies between 0.2 and 0.7 nm/h, being highest at the smallest sizes. When $C_{ORG} = 10^7$ cm$^{-3}$, GR is between 0.3 and 0.9 nm/h, increasing with size in the bins above ~2.6 nm. When $C_{ORG} = 10^8$ cm$^{-3}$, GR varies between 0.5 and 12 nm/h, and it increases with size above ~1.6 nm. When $C_{ORG} = 10^9$ cm$^{-3}$ (not shown) GR fluctuates strongly, and it obtains either extremely high (~400

nm/h) or negative values. This is due to very rapid appearance of different sized clusters in this simulation, which makes deducing GR from appearance times ambiguous. GR also fluctuates in other simulations at the smallest sizes. This results from the fact the bins at the smallest sizes contain only a few clusters, whereas the bins at the largest sizes contain hundreds of different clusters. In previous experimental studies increase of GR as a function of size has been attributed to the activation of clusters by nano-Köhler mechanism  (Kulmala et al., 2013; Tröstl et al., 2016). We study the connection between the size-

dependency of GR and nano-Köhler activation in Sect. 4.2.



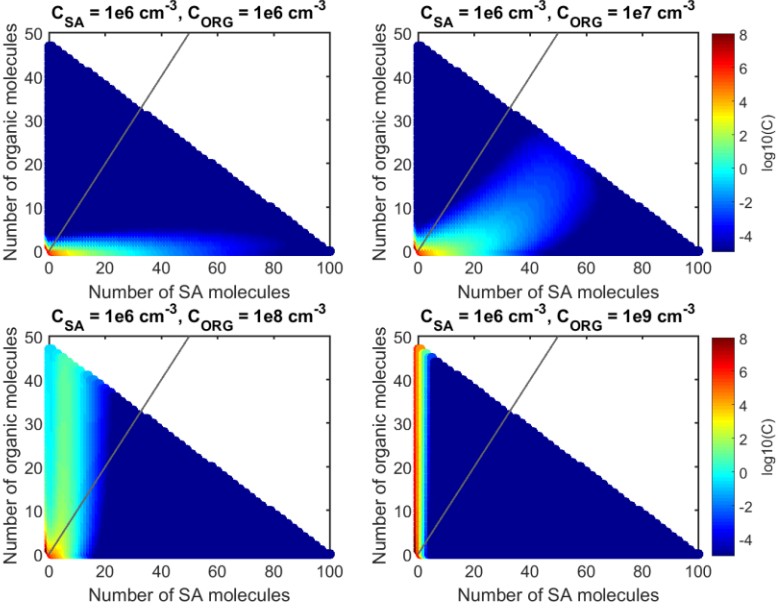

**Figure 2: Steady-state cluster distributions in simulations with SA–DMA and LVOC at $C_{SA} = 10^6$ cm$^{-3}$ and $C_{ORG} = 10^6$–$10^9$ cm$^{-3}$. The numbers of SA molecules and organic molecules in different clusters are shown on x- and y-axis, and the color indicates the concentration of a cluster.**

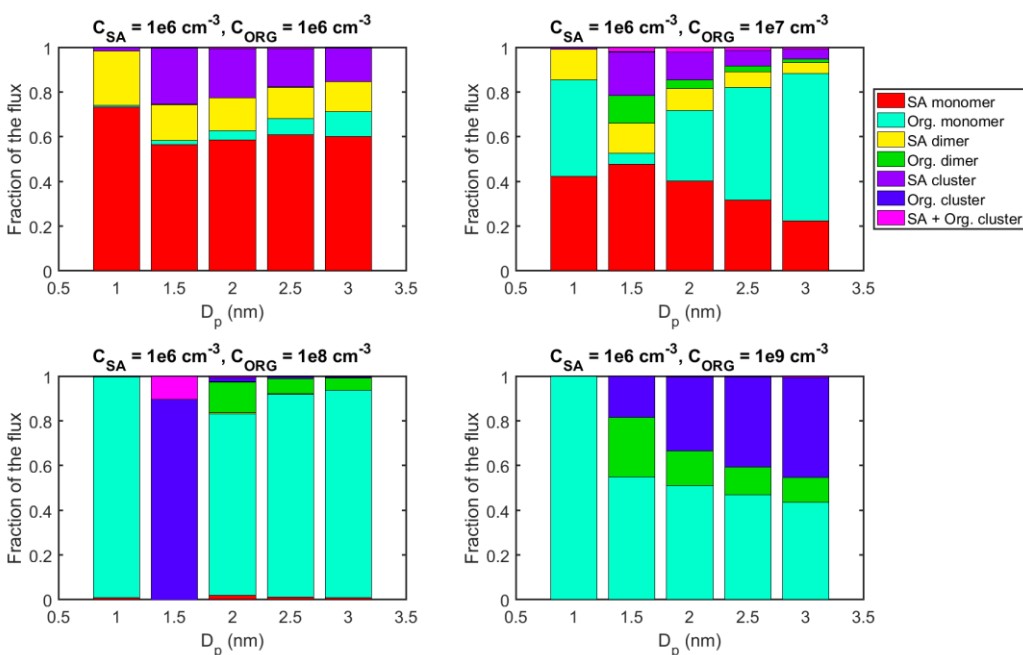

**Figure 3: Contribution of different vapor monomers and clusters to the net flux past certain threshold sizes in simulations with SA–DMA and LVOC at $C_{SA} = 10^6$ cm$^{-3}$ and $C_{ORG} = 10^6$–$10^9$ cm$^{-3}$. The values are at steady state.**





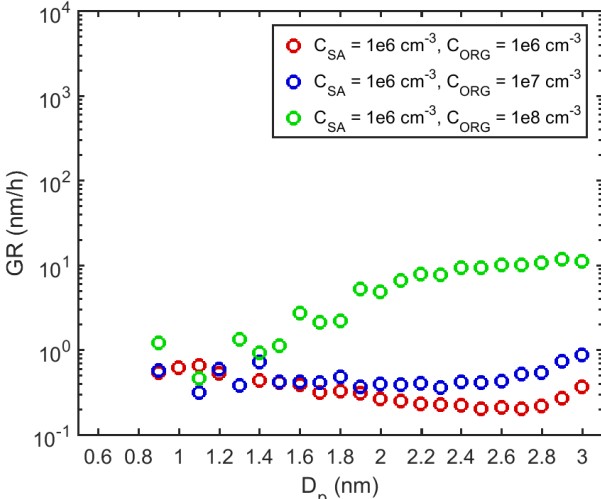

**Figure 4: Growth rates (GRs) determined for different size bins in simulations with SA–DMA and LVOC. The colors indicate vapor concentrations in each simulation.**

### 4.1.2 Simulations with ELVOC

Figures 5–7 present the results of the simulations where the model compounds were SA–DMA and ELVOC and $C_{SA} = 10^6$ cm$^{-3}$ and $C_{ORG} = 10^6$–$10^9$ cm$^{-3}$. Thus, the only difference compared to the simulations discussed in the previous section is the lower volatility ($p_{sat}$) of the organic compound.

The steady-state cluster distributions show that the organic compound contributes to the growth of the cluster population clearly more in the simulations with ELVOC than with LVOC (Fig. 5). When $C_{SA}$ and $C_{ORG}$ are $10^6$ cm$^{-3}$, SA and ELVOC

contribute to the growth approximately equally. When $C_{ORG} = 10^7$ cm$^{-3}$ or higher, the growth is already dominated by ELVOC. Consistently with the cluster distributions, the fractions of the cluster flux caused by the organic compound are clearly higher in the simulations with SA–DMA and ELVOC than in the simulations with LVOC (Fig. 6). When $C_{ORG} = 10^6$ cm$^{-3}$, SA monomer has only a rather small contribution to the fluxes at all sizes. ELVOC monomer dominates the flux at 1 nm and contributes to the fluxes significantly also at larger sizes. The clusters containing both SA and ELVOC molecules also

contribute to the fluxes above 1 nm.

When $C_{ORG} = 10^7$ cm$^{-3}$, SA monomer has only a negligible contribution to the fluxes at all sizes. ELVOC monomer dominates the flux at 1 nm. At larger sizes, the fluxes are mainly caused by ELVOC monomer, ELVOC dimer, larger ELVOC clusters and clusters containing both SA and ELVOC molecules.

In the simulations where $C_{ORG} = 10^8$ cm$^{-3}$ or $10^9$ cm$^{-3}$, the fluxes at all sizes are due to ELVOC monomer, ELVOC dimer and

ELVOC clusters. ELVOC monomer dominates the growth below 2 nm and ELVOC clusters at the larger sizes.

Overall, the difference between the cluster flux results in the simulations with ELVOC and LVOC is not only the larger contribution of the organic compound to the fluxes in the simulations with ELVOC, but also the fact that in simulations with



ELVOC, no nano-Köhler type behavior is observed. In these simulations the organic vapor can contribute to the growth at all sizes without an apparent barrier that should be overcome at small sizes.

The behavior of GR is also different in simulations with ELVOC compared to simulations with LVOC (Fig. 7). When $C_{ORG} = 10^6$ cm$^{-3}$, GR varies between 0.7 and 2.2 nm/h and reaches its highest values in the smallest size bins. When $C_{ORG} = 10^7$ cm$^{-3}$,

5    GR mostly varies between 6.5 and 9.5 nm/h, but in the smallest size bins reaches values as high as 180 nm/h. When $C_{ORG} = 10^8$ cm$^{-3}$, GR varies between 60 and 120 nm/h, without a clear size-dependency. Thus, in the simulations with ELVOC, GR does not increase with the increasing size in the same way as in the simulations with LVOC. This indicates that increasing GR observed in simulations with LVOC may be linked to the nano-Köhler type activation.

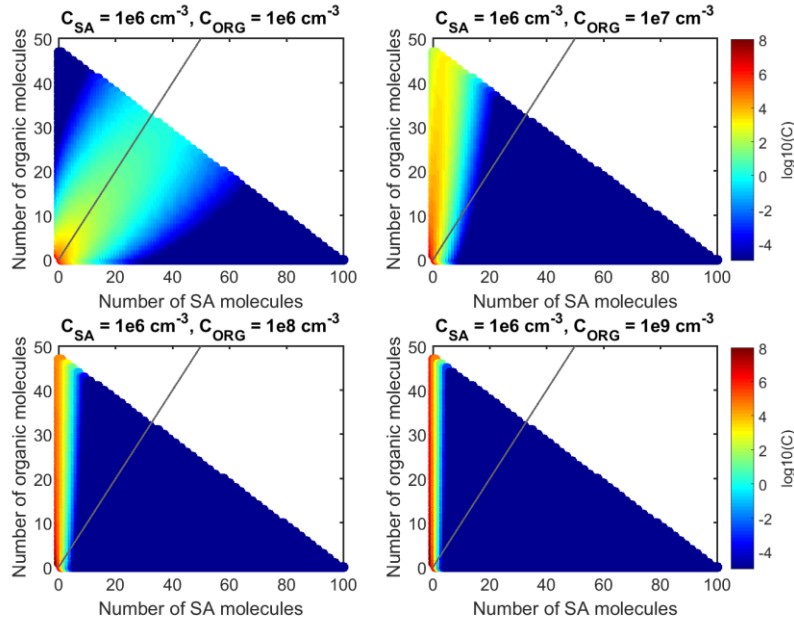

**Figure 5: Steady-state cluster distributions in simulations with SA–DMA and ELVOC at $C_{SA} = 10^6$ cm$^{-3}$ and $C_{ORG} = 10^6$–$10^9$ cm$^{-3}$. The numbers of SA molecules and organic molecules in different clusters are shown on x- and y-axis, and the color indicates the concentration of a cluster.**



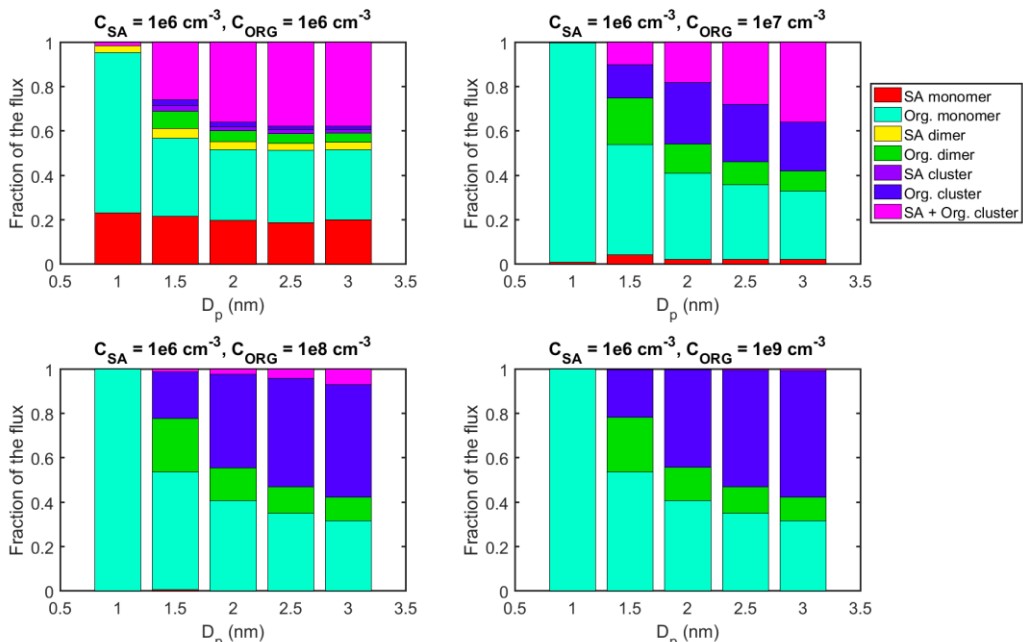

**Figure 6: Contribution of different vapor monomers and clusters to the net flux past certain threshold sizes in simulations with SA–DMA and ELVOC at $C_{SA} = 10^6$ cm$^{-3}$ and $C_{ORG} = 10^6$–$10^9$ cm$^{-3}$. The values are at steady state.**

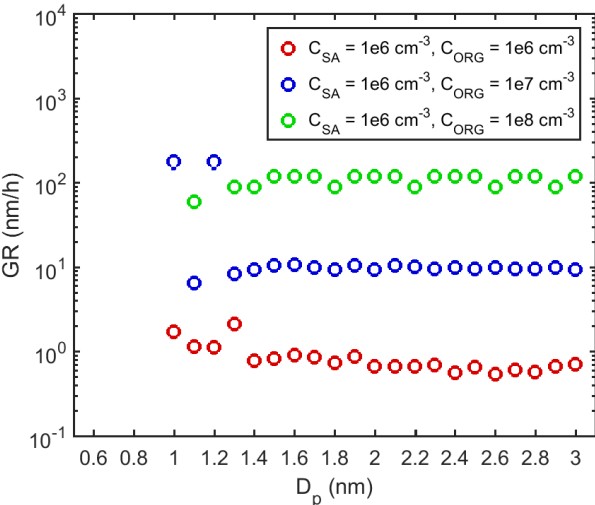

**Figure 7: Growth rates (GRs) determined for different size bins in simulations with SA–DMA and ELVOC. The colors indicate vapor concentrations in each simulation.**





### 4.1.3 Overall picture

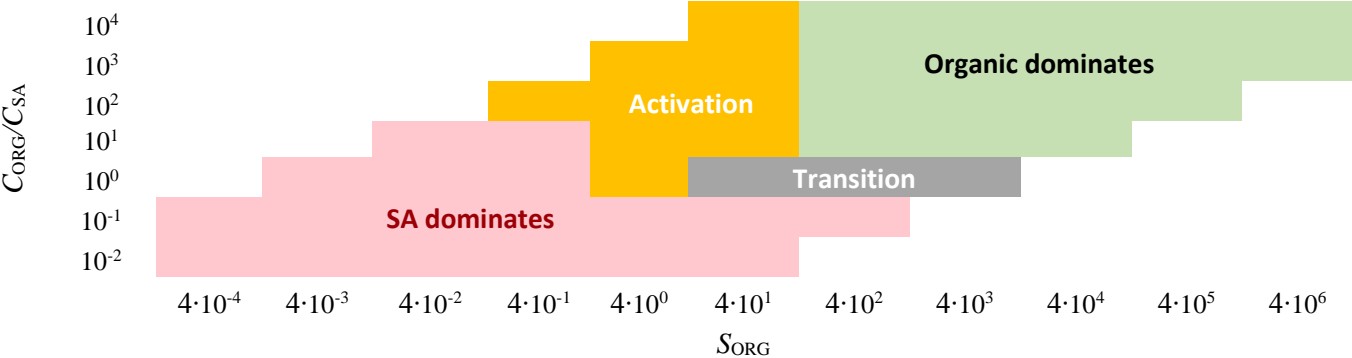

**Figure 8: Growth mechanisms with different organic vapor saturation ratios ($S_{ORG}$) and the ratios of organic vapor concentration ($C_{ORG}$) to sulfuric acid concentration ($C_{SA}$). In the red area the growth is dominated by sulfuric acid, and in the green area by the organic vapor. In the yellow and gray areas neither of them clearly dominates: in the yellow area the nano-Köhler type activation, involving an increasing contribution of the organic vapor with increasing particle size, is observed, and in the gray area there is a transition between these compounds without an activation.**

In the simulations performed with different concentrations of inorganic and organic vapors, and with a variety of organic saturation vapor pressures, different growth mechanisms can be recognized. In some simulations sulfuric acid dominates the growth of the cluster population at all sizes, whereas in other simulations the organic compound dominates. In some simulations we observe a nano-Köhler type activation, where the organic vapor starts to condense on clusters and dominate the growth after a certain size is reached. In other simulations, there is also a transition between the dominance of inorganic and organic vapor but no specific size at which the dominating compound changes.

The main variables determining which of the growth mechanism prevails in a simulation are the saturation ratio of the organic compound ($S_{ORG}$), and the ratio between the concentrations of organic and inorganic vapors ($C_{ORG}/C_{SA}$). Figure 8 illustrates the dominating growth mechanism at different values of these variables. The nano-Köhler activation is observed only in rather specific conditions: when $S_{ORG}$ = ~4–40 and $C_{ORG}/C_{SA}$ = ~10–10000. When $S_{ORG}$ and $C_{ORG}/C_{SA}$ are lower than these values, sulfuric acid dominates the growth at all sizes, while at larger values the organic vapor dominates. When $C_{ORG}/C_{SA}$ = 1 and $S_{ORG}$ is ~40 or larger, a transition without a clear activation is observed. It should be noted, though, that the activation may also occur at lower $S_{ORG}$ and $C_{ORG}/C_{SA}$ values at sizes above 3 nm, which are beyond the simulated size range. In addition, it is important to keep in mind that the conditions for different growth mechanisms depend on vapor properties and environmental conditions and thus they cannot be generalized to arbitrary compounds and conditions.

In simulations with SA–DMA and LVOC discussed in Sect. 4.1.1 (see Fig. 3), three of the different growth mechanism can be observed: the dominance of sulfuric acid (at $C_{SA}$ = $10^6$ cm$^{-3}$, $C_{ORG}$ = $10^6$ cm$^{-3}$), the dominance of organic compound (at $C_{SA}$ = $10^6$ cm$^{-3}$, $C_{ORG}$ = $10^9$ cm$^{-3}$), and activation (at $C_{SA}$ = $10^6$ cm$^{-3}$, $C_{ORG}$ = $10^7$–$10^8$ cm$^{-3}$). Similar results are also obtained in a different simulation set with the same $S_{ORG}$ and $C_{ORG}/C_{SA}$; this is illustrated in Figure A1 which shows the results of the





simulations where $p_{sat}$ of organic compound is $10^{-7}$ Pa (10 times higher than that of LVOC), $C_{SA}$ is $10^7$ cm$^{-3}$, and $C_{ORG}$ is varied between $10^7$ and $10^{10}$ cm$^{-3}$.

In the simulations with SA–DMA and ELVOC discussed in Sect. 4.1.2 (see Fig. 6), the nano-Köhler type behavior cannot be seen. At $C_{ORG} = 10^6$ cm$^{-3}$, the growth occurs both by ELVOC and SA–DMA without a clear size-dependency, and thus this

simulation represents the transition case. With higher $C_{ORG}$, ELVOC dominates the growth at all sizes.

## 4.2 Comparison between activation size in simulations and in the nano-Köhler theory

To study the cluster activation size from the simulations, we determined the contribution of different compounds to cluster flux with a higher size resolution in four simulations where nano-Köhler activation is observed (Fig. A2). We determined the size at which at least 50% of the flux is due to organic vapor monomer, which we define as the activation size $d_{sim}$ (Table 4).

To compare these results to the predictions of the nano-Köhler theory, we calculated the equilibrium saturation ratio of LVOC for different seed cluster sizes from Eq. (1). The resulting Köhler curves are illustrated in Fig. 9 and the critical diameters from the theory ($d_{theory}$) corresponding to the simulated conditions are shown in Table 4. In addition, to study the connection between accelerating growth and nano-Köhler activation, we determined the sizes at which the appearance time GR starts to significantly increase with size in each of these simulations ($d_{GR}$). This was done by finding the size at which the relative

increase of GR between two adjacent size bins is more than 10%. Note, though, that $d_{GR}$ could not be determined for the simulation with $C_{ORG} = 10^9$ cm$^{-3}$, due to strong fluctuation of GR.

**Table 4: The activation size ($d_{sim}$) determined from four simulations, the corresponding critical size from the nano-Köhler theory ($d_{theory}$), and the size at which GR starts to increase with size in each simulation ($d_{GR}$). The activation size is here defined as the size**
**when 50% of the cluster flux is due to organic vapor monomer. The concentrations of sulfuric acid ($C_{SA}$) and organic vapor ($C_{ORG}$), their ratio, organic saturation vapor pressure ($p_{sat,ORG}$) and organic saturation ratio ($S_{ORG}$) are shown for each case.**

| $C_{SA}$ (cm$^{-3}$) | $C_{ORG}$ (cm$^{-3}$) | $C_{ORG}/C_{SA}$ | $p_{sat,ORG}$ (Pa) | $S_{ORG}$ | $d_{sim}$ (nm) | $d_{theory}$ (nm) | $d_{GR}$ (nm) |
|---|---|---|---|---|---|---|---|
| $10^6$ | $10^7$ | 10 | $10^{-8}$ | 3.8 | 2.5 | 4.1 | 2.7 |
| $10^6$ | $10^8$ | 100 | $10^{-8}$ | 38.0 | 1.75 | 1.7 | 1.7 |
| $10^6$ | $10^8$ | 100 | $10^{-7}$ | 3.8 | 2.75 | 4.1 | 2.4 |
| $10^6$ | $10^9$ | 1000 | $10^{-7}$ | 38.0 | 2.0 | 1.7 | - |





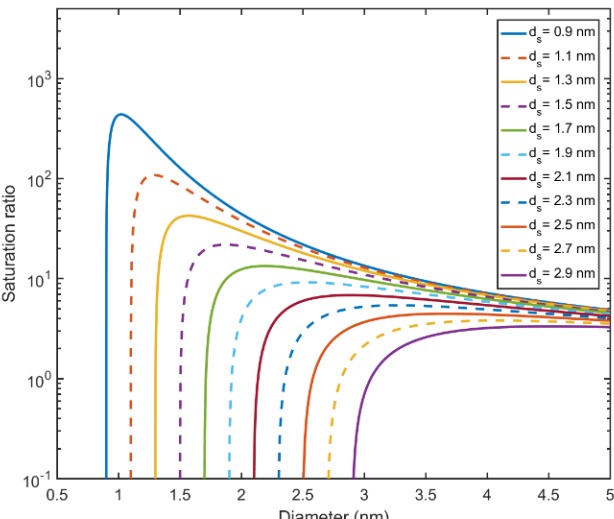

**Figure 9: Köhler curves showing the equilibrium particle size at different saturation ratios for LVOC. The line color shows the diameter of the seed particle ($d_s$). The maxima of the curves correspond to the critical sizes.**

In the simulations where $S_{ORG} = 38$, $d_{sim}$ and $d_{theory}$ are rather close to each other: the difference between them is 0.05–0.3 nm

depending on vapor concentrations. However, when $S_{ORG} = 3.8$, $d_{sim}$ is clearly lower (>1 nm) than $d_{theory}$. This discrepancy is likely related to the fact that the theory does not account for the stochastic collisions between inorganic and organic vapor molecules and clusters, which can result in the growth of the clusters by organic vapor already before the theoretical critical size (see also Sect. 2). Thus, although nano-Köhler –type behavior occurs in the simulated systems, the theory cannot be used to predict the exact size at which the organic vapor starts to dominate the cluster growth.

On the other hand, the difference between $d_{sim}$ and $d_{GR}$ is only 0.05–0.35 nm in different simulations. This suggests that, at least in some conditions, the increase in particle GR may indicate that the organic vapor starts to condense on the clusters around that size. However, as the behavior of GR is also affected by the variation of vapor concentrations and other conditions, this does not necessarily apply in real atmospheric systems (see Sect. 4.3.2).

It should finally be noted that it is not entirely obvious how $d_{sim}$ should be defined. In some simulations there is a distinct size

after which most of the flux is due to organic vapor monomer, but in some simulations the change is more gradual. In addition, in many simulations organic dimer or larger organic clusters contribute to the growth significantly at sizes where the net flux of the organic vapor monomer is negative. Thus, in these cases the growth is governed by the organic compound already before the activation size, which we determine only based on the contribution of the organic monomer.



### 4.3 Sensitivity of the results to vapor properties and time-dependency of vapor concentrations

#### 4.3.1 Effect of inorganic vapor properties and organic compound mass

To study the effect of inorganic vapor properties on the growth of the cluster population, we performed additional simulations where the organic compound was LVOC and the inorganic compound was SA–NH$_3$, which has a non-zero saturation vapor
pressure ($10^{-9}$ Pa) and a slightly smaller mass than SA–DMA (see Table 2). In these simulations the contribution of inorganic compound to the cluster fluxes, especially that of vapor monomer, is smaller than in the simulations with SA–DMA, due to higher evaporation flux (Fig. A3). The GRs determined for different size bins are generally slightly lower in the simulations with SA–NH$_3$ than with SA–DMA (Fig. A4). However, when $C_{SA} = 10^6$ cm$^{-3}$ and $C_{ORG} = 10^6$ cm$^{-3}$, GR at sizes above 2.5 nm is higher in the simulation with SA–NH$_3$ than with SA-DMA. In this case the GRs determined based on the appearance times
seem not to represent the magnitude of collision-evaporation fluxes which are larger in the simulation with SA–DMA.
We also performed simulations with SA–DMA and an organic compound with the same saturation vapor pressure as LVOC ($10^{-8}$ Pa) but two times larger mass (LVOC$_{large}$). In the simulations where $C_{SA} = 10^6$ cm$^{-3}$, and $C_{ORG} = 10^7$–$10^9$ cm$^{-3}$, the organic compound was found to contribute to the growth clearly less than in the simulations with LVOC (Fig. A5). The main reason for the difference is that the evaporation rate is higher for the compound with a higher mass, as the molecular mass is located
inside the exponent in the Kelvin formula (Eq. 3). In addition, because the diameter of LVOC$_{large}$ monomer is higher than 1 nm, it cannot contribute to the flux at 1 nm by default. Consistently with the lower contribution of the organic compound to the growth, the apparent cluster GRs are also lower in the simulations with LVOC$_{large}$ than with LVOC (Fig A6). Overall, these results demonstrate that the properties of both inorganic and organic compounds, including saturation vapor pressure but also other properties, can significantly affect the contribution of different compounds to the cluster growth. Therefore, our
simulation results regarding, for example, the conditions for different growth mechanisms (Sect. 4.1.3) cannot be generalized for arbitrary compounds.

#### 4.3.2 Effect of time-dependency of vapor concentrations

To study the effect of time-dependent vapor concentrations, two additional simulations were performed using SA–DMA and LVOC as model compounds. In the first simulation the behavior of both vapor source rates was set to resemble the diurnal
cycle of atmospheric sulfuric acid concentration (Petäjä et al., 2009). In the second simulation the source rate of SA–DMA had a similar behavior but the source rate of LVOC was set constant. The resulting vapor concentrations and the GRs determined for different size bins are shown in Fig. 10. The time-evolution of the net flux caused by different vapor monomers and clusters past different sizes in these simulations is shown in Fig. A7. Based on the flux results, in both of these simulations the organic vapor starts to significantly contribute to cluster growth around 2 nm, which corresponds to nano-Köhler type
activation. In the simulation where both $C_{SA}$ and $C_{ORG}$ have maxima, the activation can also be seen as an increase in GR at sizes above 2 nm. However, in the simulation with constant $C_{ORG}$, GR increases with size almost linearly, without any signs





of activation. This shows that the size-dependency of GR is affected by dynamic variations in the system, and therefore GR alone cannot be used to deduce the cluster growth mechanism.

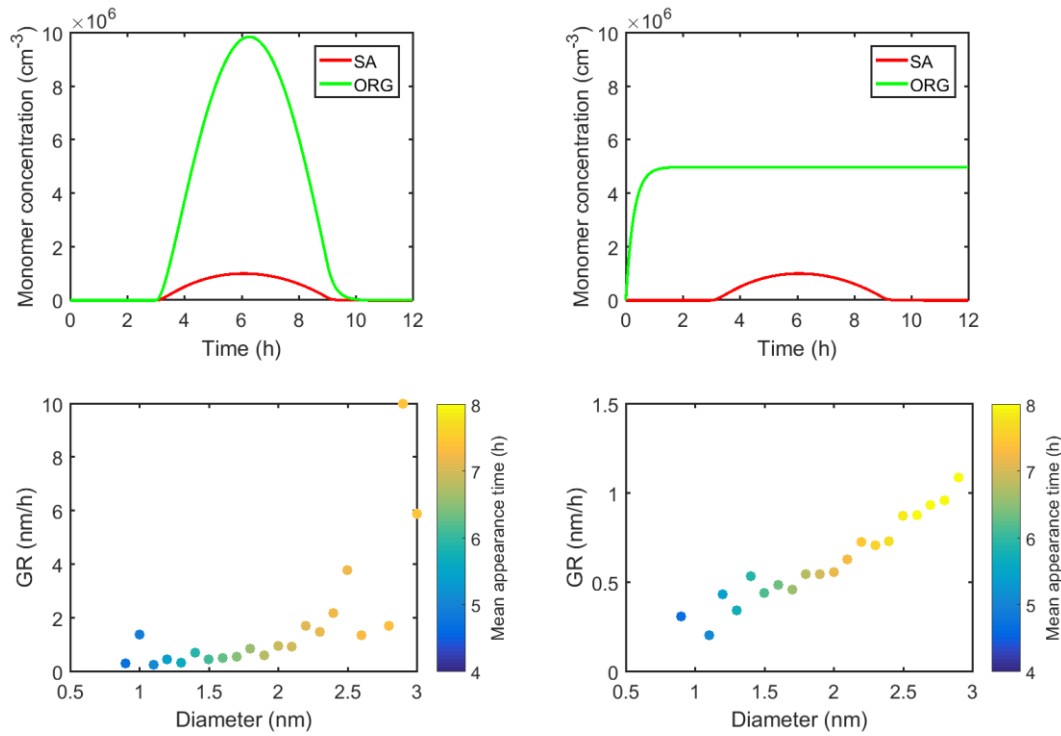

**Figure 10: The concentrations of SA–DMA and LVOC monomers in the simulations where vapor source rates are time-dependent (upper panel), and growth rates (GRs) determined for different size bins in these simulation (lower panel). In the lower panel, the colors show the average of the appearance times of the adjacent size bins used for determining GRs.**

## 5 Conclusions

Recent experimental results indicate that organic vapors can participate in the growth of nanometer-sized atmospheric molecular clusters. One of the mechanisms proposed to depict this process is the nano-Köhler theory, which describes the activation of inorganic clusters to growth by a soluble organic vapor. However, it is unclear how well the simple theory is able to describe the dynamics of real molecular systems with a distribution of clusters of varying size and composition and time-dependent environmental conditions.

In this work we studied the capability of the nano-Köhler theory to describe the growth of atmospheric clusters by simulating the dynamics of a cluster population in the presence of a sulfuric acid–base mixture and an oxidized organic compound. We found that nano-Köhler type behavior occurs when the saturation ratio $S_{ORG}$ of the organic vapor and the ratio $C_{ORG}/C_{SA}$ between organic and inorganic vapor concentrations are in the correct range, namely $C_{ORG}/C_{SA} = \sim10{-}10000$ and $S_{ORG} = \sim4{-}$





40. When $S_{ORG}$ and $C_{ORG}/C_{SA}$ are lower than these values, sulfuric acid dominates the growth at the studied sizes, whereas with the larger values the organic compound dominates. The conditions in which we observe nano-Köhler type behavior can also occur in the atmosphere: in boreal forests $C_{SA}$ is typically $\sim 10^6$ cm$^{-3}$ during NPF events (Petäjä et al., 2009), and $C_{ORG}$ is likely between $\sim 10^5$ and $10^8$ cm$^{-3}$ (Jokinen et al., 2017), which would result in $S_{ORG} = \sim 0.04$–40 with $p_{sat,ORG} = 10^{-8}$ Pa and $S_{ORG} = \sim 40$–$40 \cdot 10^3$ with $p_{sat,ORG} = 10^{-11}$.

Although nano-Köhler type activation occurs in our simulations, the nano-Köhler theory is too simple to predict the exact size at which the organic vapor starts to significantly condense on the clusters. This observation demonstrates that the growth of sub-3 nm clusters can be described accurately only if the whole cluster distribution with a variety of different cluster composition and sizes is taken into account. Especially when vapor saturation ratios are high, cluster–cluster collisions contribute substantially to the growth in addition to vapor monomers.

Finally, we also observed that in the simulations with nano-Köhler type behavior, apparent cluster GRs start to increase close to the size at which the organic vapor starts to dominate the growth. However, such behavior of GR may also be due to other dynamic processes, such as varying vapor concentrations, and therefore one should not make conclusions about the cluster growth mechanism solely based on GR. To accurately determine the initial growth mechanisms and the participating compounds would ideally require (1) quantitative measurements of the composition of sub-3 nm clusters, and (2) improved understanding of cluster thermodynamics, including composition and size-dependent evaporation rates. Combined with cluster population modeling, such assessments will be able to provide a more comprehensive understanding of the clustering processes.

**Code and data availability**

Model simulations were performed using Atmospheric Cluster Dynamics Code (ACDC). The code and the simulated data are available upon request from the authors.





## Appendix

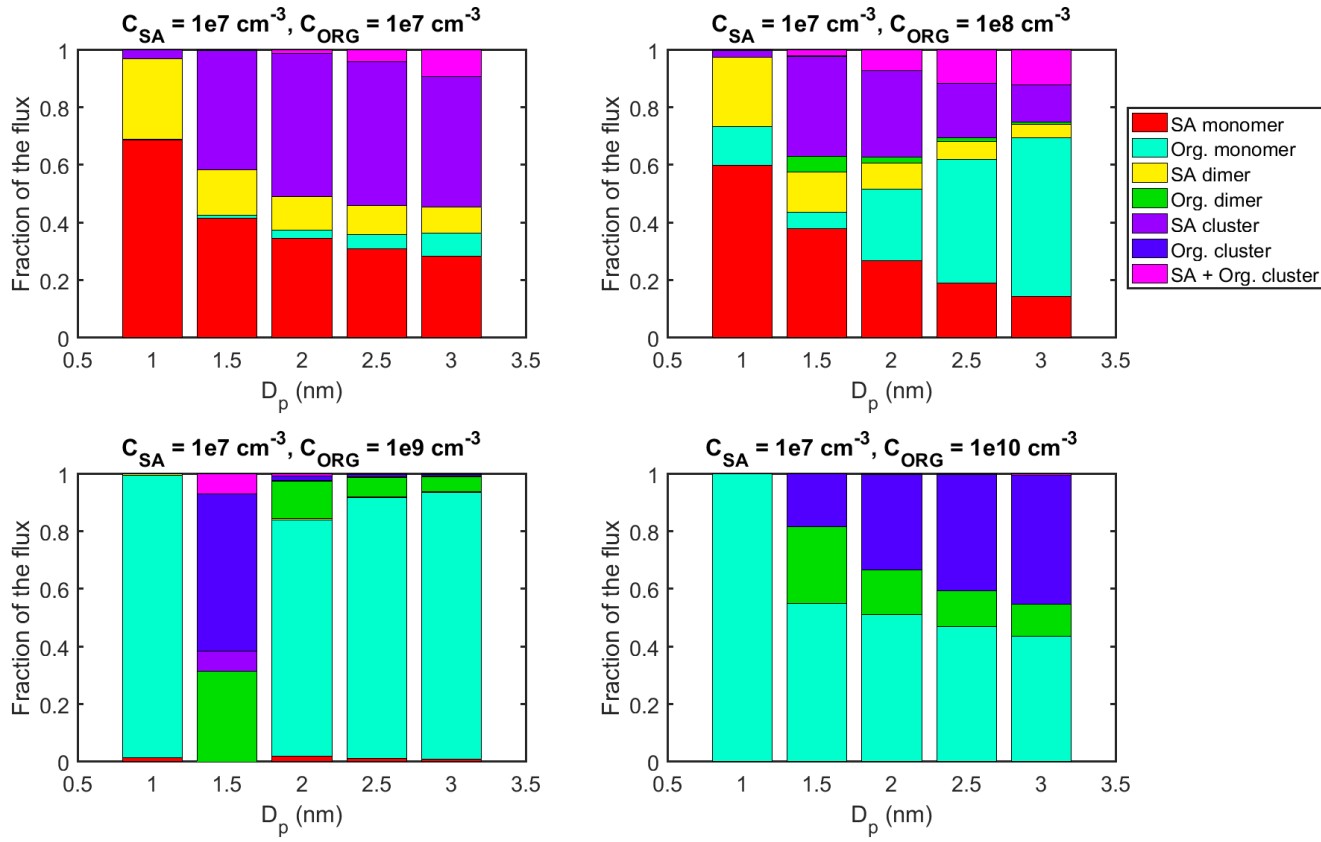

**Figure A1: Contribution of different vapor monomers and clusters to the net flux past certain threshold sizes in simulations with SA–DMA and an organic compound (ORG) with $p_{sat} = 10^{-7}$ Pa at $C_{SA} = 10^7$ cm$^{-3}$ and $C_{ORG} = 10^7$–$10^{10}$ cm$^{-3}$. The values are at steady state.**



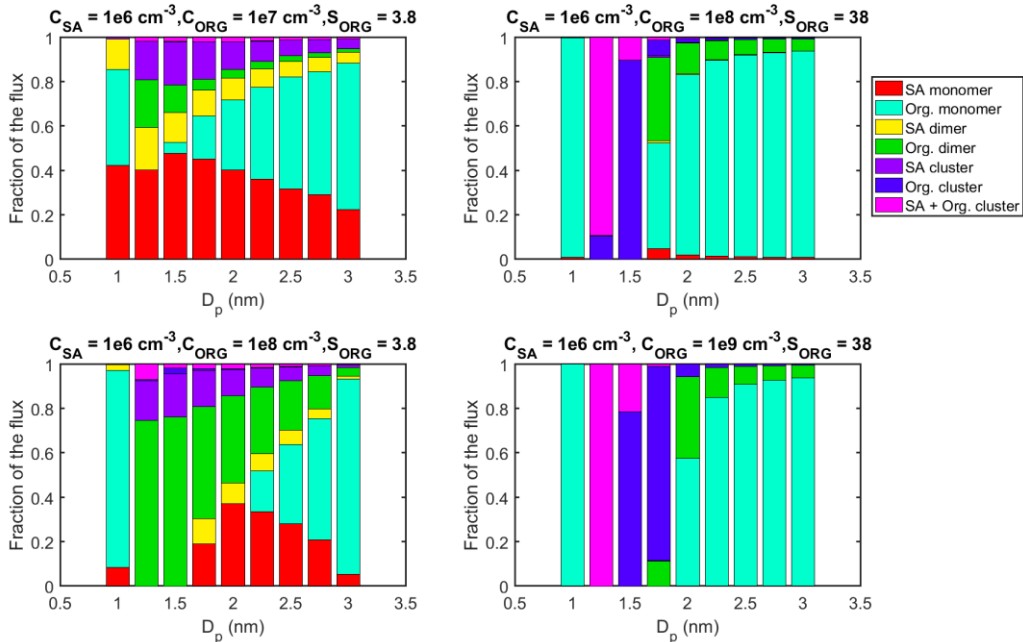

**Figure A2: Contribution of different vapor monomers and clusters to the net flux past different threshold sizes in simulations with SA–DMA and an organic compound (ORG). The vapor concentrations and the saturation ratio of organic compound ($S_{ORG}$) are shown above the figures for each simulation. The values are at steady state.**

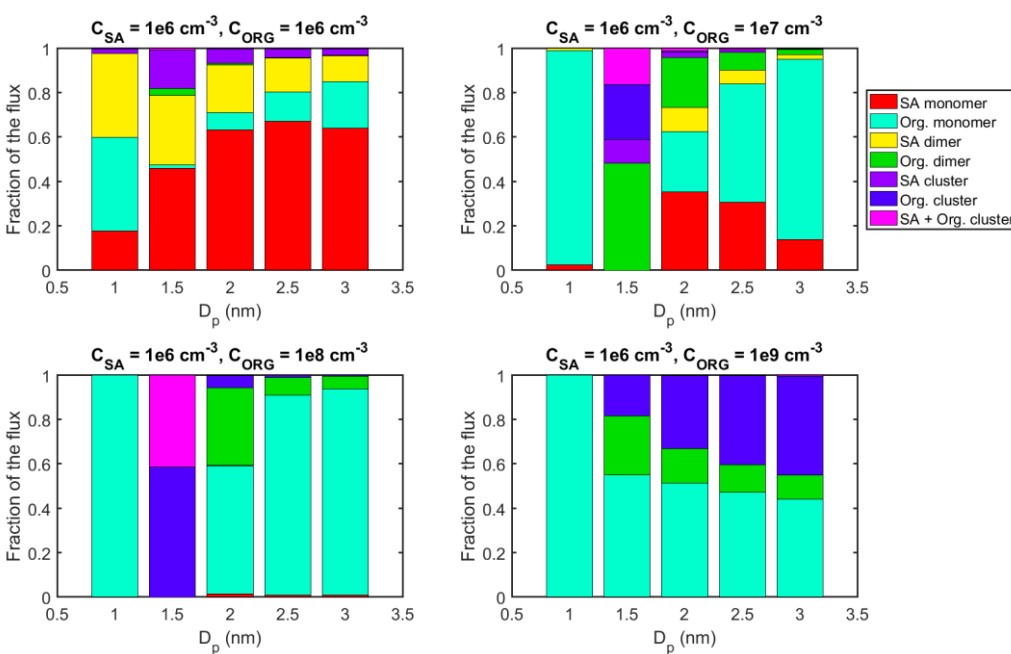

**Figure A3: Contribution of different vapor monomers and clusters to the net flux past different threshold sizes in simulations with SA–NH$_3$ and LVOC at $C_{SA} = 10^6$ cm$^{-3}$ and $C_{ORG} = 10^6$–$10^9$ cm$^{-3}$. The values are at steady state.**

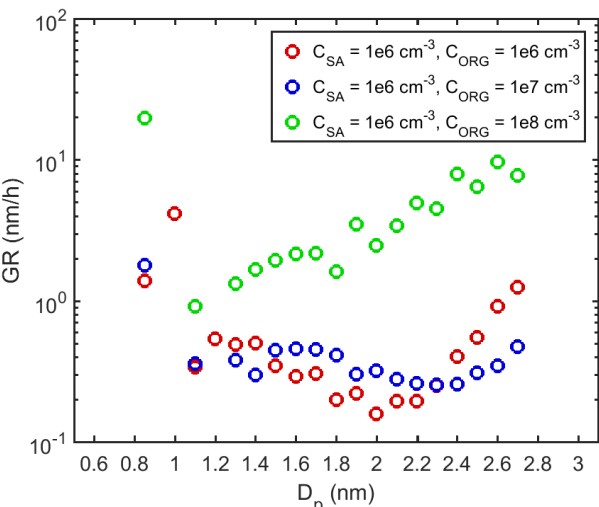

**Figure A4: Growth rates (GRs) determined for different size bins in simulations with SA–NH₃ and LVOC. The colors indicate vapor concentrations in each simulation.**

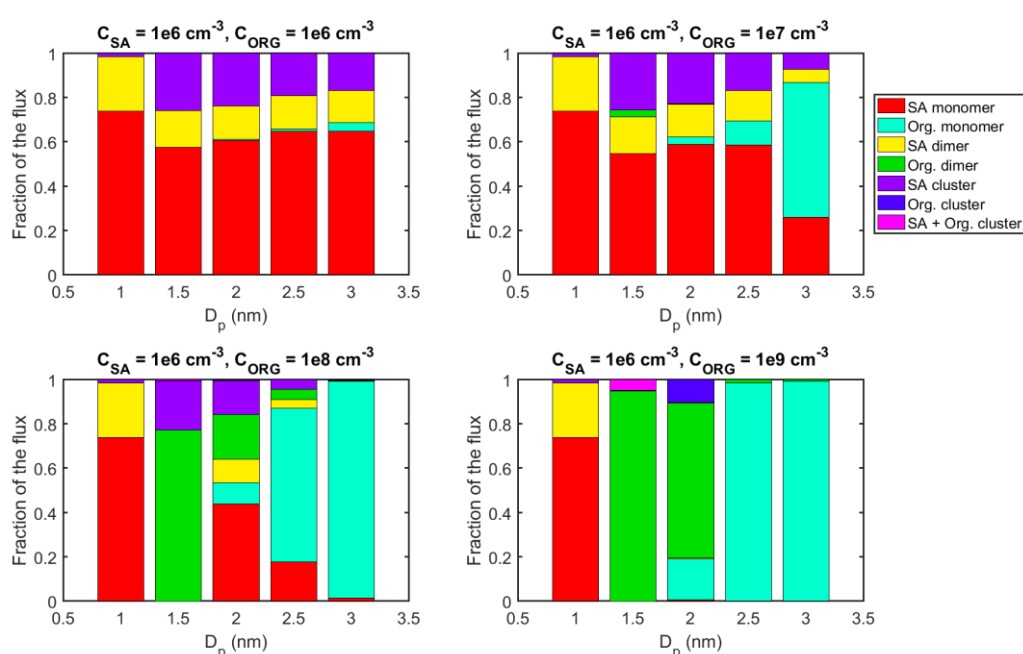

**Figure A5: Contribution of different vapor monomers and clusters to the net flux past certain threshold sizes in simulations with SA–DMA and an organic compound with $p_{sat} = 10^{-8}$ Pa and $m = 600$ amu (LVOC$_{large}$) at $C_{SA} = 10^6$ cm$^{-3}$ and $C_{ORG} = 10^6$–$10^9$ cm$^{-3}$. The values are at steady state.**



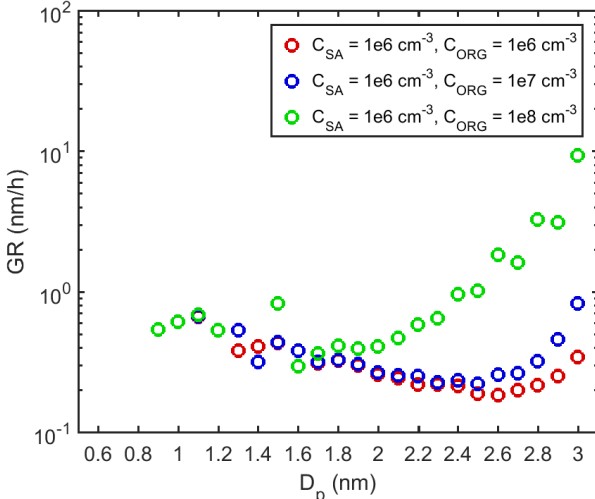

**Figure A6: Growth rates (GRs) determined for different size bins in simulations with SA–DMA and an organic compound with $p_{sat}$ = $10^{-8}$ Pa and $m$ = 600 amu (LVOC$_{large}$) at $C_{SA}$ = $10^6$ cm$^{-3}$ and $C_{ORG}$ = $10^6$–$10^9$ cm$^{-3}$. The colors indicate vapor concentrations in each simulation.**



**Figure A7: The net flux caused by different vapor monomers and clusters past certain threshold sizes in the simulations with time-dependent concentrations of SA–DMA and LVOC. The left and the right panels correspond to the left and the right panels in Fig. 10.**





**Author contribution**

JK, TO, IR and MK planned the model simulations and JK performed them. JK prepared the manuscript with contributions from all co-authors.

**Competing interests**

The authors declare that they have no conflict of interest.

**Acknowledgements**

This study was funded by the European Research Council (grant no. 742206), the Academy of Finland Center of Excellence programme (grant no. 307331), Swedish Research Council Formas (grant no. 2015-749), and Knut and Alice Wallenberg foundation (academy fellowship AtmoRemove).

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
