# Peer review of "Exploring the potential of the nano-Köhler theory to describe the growth of atmospheric molecular clusters by organic vapors"

_Atmospheric Chemistry and Physics, 2018_

## Referee Comment (RC1) · Anonymous Referee #1 · 3 Jul 2018

**Comments on "Exploring the potential of the nano-Köhler theory to describe the growth of atmospheric molecular clusters by organic vapors" by Jenni Kontkanen et al.**

July 3, 2018

**General comments**

Manuscript explores the validity of the so-called nano-Köhler theory to describe the on-set of the growth of atmospheric particles. Although no simple answer to the question posed by the title is given, the manuscript provides a commendable effort in focusing on the validity of simplified assumptions often used—and too often overlooked—when modelling atmospheric cluster and particle processes; in this sense, the manuscript provides a natural continuation of the work performed earlier by some of the authors using the same methodology (ACDC cluster population model; Olenius and Riipinen, 2017). The presentation is clear and conclusions follow logically from the computational results (see, however, specific comments), although as a non-native English speaker I feel that usage of some additional commas could improve the representation.

However, even more (theoretical) insight could be obtained from the presented computational results, and the authors should consider including more detailed discussion on the manuscript.

- The premise of the nano-Köhler theory is that homogeneous nucleation of inorganic clusters is followed by activation of the same clusters by organic vapour condensation, while the results shown indicate that in actual atmospheric conditions the situation may not be so straightforward. In earlier studies using the same methodology (ACDC), the nature of the first step—formation of sulphuric acid–ammonia/amine clusters—has been found spontaneous, i.e. posing no thermodynamic

barrier, under some atmospherically relevant conditions. Likewise, results given in the manuscript for the ELVOCs (Sect. 4.1.2) seem to indicate barrierless condensation of organic vapour. Thus, it seems that there are four possible scenarios: i) thermodynamic barrier for both inorganic cluster formation (nucleation) and organic condensation (activation), ii) thermodynamic barrier inorganic cluster formation and barrierless condensation, iii) no thermodynamic barrier for inorganic cluster formation but thermodynamic barrier for organic condensation, and iv) no thermodynamic barrier for inorganic cluster formation nor organic condensation. It would be interesting to know how the number of thermodynamic barriers would contrast to the overall picture presented in Fig. 8.

- Related to the overall picture and schematics of Fig. 8, authors remark that "conditions for different growth mechanisms depend on vapor properties and environmental conditions and thus they cannot be generalized to arbitrary compounds and conditions". However, as authors have performed simulations using two different scenarios for particle loss with different size dependencies, it would be interesting to know how sensitive this scheme is to the nature and strength of particle losses.

- Table 3 gives a good summary on different simulation sets, and it would help the reader if these would also be referred accordingly in Results and discussion.

**Specific comments and technical corrections**

- Page 1, line 27: 'aerosol forcing' → 'aerosol radiative forcing'.

- Page 2, lines 1–12: The role of ions for the NPF process in addition to organic compounds and bases could be mentioned. Related to this, there is no reference corresponding to 'Kirkby et al., 2016' in the list of references.

- Starting from page 2, there are several references to articles 'Kulmala, 2004' and 'Kulmala et al., 2004' in the manuscript. However, only the one corresponding to 'Kulmala, 2004' is given in the list of references, although it should be 'Kulmala et al., 2004'. The list of references has also other issues and should be revised by the authors.

- Page 3, line 11: Heterogeneous, not homogeneous, nucleation of the organic vapour should be implied.

- In relation to Eq. (1), $a_{\mathrm{org}}$ is used to denote the activity coefficient of the organic compound. As $a$ is commonly used for the activity, this seems somewhat misleading. I would recommend using $f_{\mathrm{org}}$ for the activity coefficient, as $\gamma$ has been already reserved for other use. Also, the surface tension in the Kelvin term should refer to the droplet/cluster as whole, not to the organic compound.

- Page 4, lines 17–18: It should be noted that if adsorption of vapour on the insoluble seed surface is taken into account, it is possible to have a maximum in the saturation ratio vs. cluster size curve [1].

- Page 4, lines 18–30: Although mainly phrased in terms of water vapour, theoretical and simulation results can be found from the literature focusing on the nucleation/activation-transition [3, 4, 5], some of which might be relevant for discussion here.

- Figure 1: This is a very good figure illustrating the differences between simplified nano-Köhler theory and the real system behaviour. However, the meaning of double-headed thin arrow in the real system description is not clear, does it imply forward and backward crossing of the thermodynamic barrier?

- Table 1: Is there any reason, why the condensing organic vapour has to be water-soluble in the nano-Köhler theory?

- Page 6, line 8: General Dynamic Equation (not Dynamics). Also, although this is a matter of taste, Eq. (2), when given in molecular resolution, could be referred as an extended Smoluchowski coagulation equation, considering Marian Smoluchowski's seminal contribution to the theory.

- Page 6, line 15: An original reference [2] for the ACDC model should be given.

- Equation (3): Although containing the Kelvin term, this equation could be more properly referred as a condition of detailed balance than the Kelvin formula.

- Page 7, line 16: Mass of 500 amu is given here for the $\mathrm{LVOC_{large}}$, while in Table 2 and caption of Fig. A6, 600 amu are given. Which one is right?

- Page 7, lines 26–29: Would there be other likely contributions, besides Raoult's law effect for the organic vapour, from the inclusion of water vapour into simulations?

- Table 3 and page 21, line 5: The unit for pressure (Pa) is missing.

- Figures 2 and 5: Does the solid line indicate clusters with 1:1 stoichiometry? I could not find any explanation from the text.

- Page 19, lines 13–15: When considering Eq. (3), this is right when considering a given compound. However, as in general higher molecular mass implies smaller equilibrium vapour pressure, this statement sounds odd. It should be noted that the ratio $m_{org}/\rho_{org}$ in Eq. (3) refers to the (partial) molecular volume of the organic compound in the cluster, correlating strongly with the surface area of a (spherical) molecule at the surface. From this perspective, it might be better to rephrase the sentence pointing out the importance of molecular volume/exposed surface area instead of molecular mass on the equilibrium vapour pressure over a curved surface.

**Additional references**

[1] A. Laaksonen and J. Malila, Atmos. Chem. Phys., 16, 135–143, 2016.

[2] M. J. McGrath, T. Olenius, I. K. Ortega, V. Loukonen, P. Paasonen, T. Kurtén, M. Kulmala, and H. Vehkamäki, Atmos. Chem. Phys., 12, 2345–2355, 2012.

[3] P. Mirabel, H. Reiss, and R. K. Bowles, J. Chem. Phys., 113, 8194–8199, 2000.

[4] H. Reiss and G. J. M. Koper, J. Phys. Chem., 99, 7837–7844, 1995.

[5] R. P. Sear, Europhys. Lett., 83, 66002, 2008.

---

## Referee Comment (RC2) · Anonymous Referee #2 · 9 Jul 2018

"Exploring the potential of the nano-Köhler theory to describe the growth of atmospheric molecular clusters by organic vapors," by Kontkanen et al., explores the conditions in which nano-Kohler theory can be applied to the initial stages of new particle formation (NPF). The development of simple models that accurately represent NPF and the subsequent growth of nanometer-sized particles are needed in order to assess the importance of NPF in climate and air quality. This study can potentially address these needs, and therefore is quite appropriate for publication in ACP. I do, however, have one major concern that the authors should address before recommending publication. In addition, I will recommend a number of minor corrections.

[Figure]

This study has, as its main objective, the determination of whether nano-Kohler theory may be appropriate for representing NPF for range of compounds (H2SO4, LVOC, and ELVOC) and concentrations (1E6 – 1E8) that are representative of ambient air in many locales. In order to test their implementation of nano-Kohler, the authors compared their results to those of a cluster kinetics model. Herein lies my concern. Since the authors use their comparisons between nano Kohler and cluster kinetics models as their metric for whether nano-Kohler is appropriate for describing atmospheric NPF, this paper should be more appropriately titled "Exploring the potential of the nano-Köhler theory to describe the growth of atmospheric molecular clusters by organic vapors as predicted by a cluster kinetics model." I assume that the authors wish the readers to interpret these results more generally, i.e., associate the information shown in Figure 8 (which, as an aside, is a wonderful graphic!) with actual atmospheric concentrations of H2SO4 and organics. But this is not what's being tested, nor have the authors placed effort into convincing the reader that the assumptions made in implementing the cluster kinetics mode actually result in an accurate description of atmospheric NPF.

I see two possible ways to address this issue, both of which could ideally be applied to this study. The first is to address my concern about the accuracy of cluster kinetics modeling for describing NPF under the range of conditions that are the foci of this study. Rather than assuming that the reader interprets the results of cluster kinetics modeling as"truth," the authors need to provide clear evidence of this fact. This includes the validity of the various assumptions used in that model, such as hard-sphere collisions, evaporation rates using Kelvin Theory, etc.

My other recommendation is to use experimental data to compare to the results of nano-Kohler. Prior studies have explored cluster growth rates as a function of mea-sured H2SO4 concentrations (e.g., "Size and time-resolved growth rate measurements of 1 to 5 nm freshly formed atmospheric nuclei," Kuang et al., ACP, 2012), so it would seem a simple task to take measured growth rates and [H2SO4] and explore the predicted growth rate from nano-Kohler using realistic assumptions for [LVOC] and
[ELVOC].

My first recommendation, I feel, is necessary for this paper. My second recommendation would allow readers to have a lot more confidence that the data shown in Figure 8 is truly representative of the real atmosphere.

The following are minor suggestions, where each comment is preceded by the page and line number.

P1, L29: Shouldn't the word "including" be replaced by "specifically"? Including suggests that the phrase that follows is a process that differs from NPF, but in my view it specifically defines NPF. In general I would recommend to the authors that they do a better job of defining, very early in the manuscript, what is meant by NPF. In this paper, the focus is on the formation of the cluster and the growth up to a few nanometers in diameter. One gets that point later in the paper, but I feel it could be made more clear from the start (this includes the abstract).

P3, L21: "to study in what kind" is awkward phraseology. I suggest "to study the conditions under which"

P3, L26: "The nano-Kohler" does not require the article "The" . . . this is a common error throughout the manuscript.

P4, L10: The term "seed cluster" is used here but it really hasn't been introduced. What is a seed cluster and why is it required?

P6, L27: This stated loss rate due to dilution is unique to the CLOUD experiments, as it depends on the flow rates into and out of the chamber, That authors should state this fact.

P7, L26: Wouldn't adding water content to the model also increase uptake of some compounds such as H2SO4 and other hygroscopic organics, due to increased surface area?

---

## Author Comment (AC1) · 23 Aug 2018

**REPLIES TO REFEREES**

We thank the referees for their insightful comments and suggestions that have helped us to improve our manuscript.

We have answered to each of the referee's comments below. The reviewers' comments are shown in **bold**, and the text that has been added to, or modified in, the revised manuscript is shown in *italics*. The page and line numbers given in the answers refer to those in the ACPD version of the manuscript.

**Reply to Referee #1**

**General comments**

**Manuscript explores the validity of the so-called nano-Köhler theory to describe the on-set of the growth of atmospheric particles. Although no simple answer to the question posed by the title is given, the manuscript provides a commendable effort in focusing on the validity of simplified assumptions often used—and too often overlooked—when modelling atmospheric cluster and particle processes; in this sense, the manuscript provides a natural continuation of the work performed earlier by some of the authors using the same methodology (ACDC cluster population model; Olenius and Riipinen, 2017). The presentation is clear and conclusions follow logically from the computational results (see, however, specific comments), although as a non-native English speaker I feel that usage of some additional commas could improve the representation. However, even more (theoretical) insight could be obtained from the presented computational results, and the authors should consider including more detailed discussion on the manuscript.**

**The premise of the nano-Köhler theory is that homogeneous nucleation of inorganic clusters is followed by activation of the same clusters by organic vapour condensation, while the results shown indicate that in actual atmospheric conditions the situation may not be so straightforward. In earlier studies using the same methodology (ACDC), the nature of the first step—formation of sulphuric acid–ammonia/amine clusters—has been found spontaneous, i.e. posing no thermodynamic barrier, under some atmospherically relevant conditions. Likewise, results given in the manuscript for the ELVOCs (Sect. 4.1.2) seem to indicate barrierless condensation of organic vapour. Thus, it seems that there are four possible scenarios: i) thermodynamic barrier for both inorganic cluster formation (nucleation) and organic condensation (activation), ii) thermodynamic barrier inorganic cluster formation and barrierless condensation, iii) no thermodynamic barrier for inorganic cluster formation but thermodynamic barrier for organic condensation, and iv) no thermodynamic barrier for inorganic cluster formation nor organic condensation. It would be interesting to know how the number of thermodynamic barriers would contrast to the overall picture presented in Fig. 8.**

The existence of a thermodynamic barrier depends mainly on vapor saturation ratio, which is determined by vapor concentration and its saturation vapor pressure over the surface of a cluster or a particle. Therefore, the region on the right-hand side of Fig. 8, where organic vapor dominates the growth at all sizes, corresponds to the situation where there is no barrier, or a relatively small barrier, for organic vapor. The regions where nano-Köhler type behaviour is observed or sulfuric acid dominates the growth correspond to the situation with a barrier for the organic vapor. Correspondingly, the region on the left-hand side corresponds to the situation where there is no barrier, or only a small barrier, for sulfuric acid. In other words, Fig. 8 can be interpreted to give information on the relative barriers: the compound that has no barrier or a significantly lower barrier to cluster and condense is likely to dominate the whole formation process. Activation or transition

occurs when one of the compounds overcomes its thermodynamic barrier, regardless of if the clustering of the other compound driving the initial cluster formation involves barriers or not. It must be noted that activation/transition is in fact the only case that always involves a thermodynamic barrier: cases where one of the compounds dominates can also be due to differences in the vapor-phase concentrations, and do not necessarily involve thermodynamic barriers (i.e. a compound with a high concentration dominates). To clarify this in the manuscript we added the following sentence in the end of the Sect. 4.1.3:

*On a general level, Fig. 8 can be interpreted to give information on the relative thermodynamic barriers of inorganic and organic compounds: the compound that has no barrier or has a significantly lower barrier to cluster and condense than the other compound is likely to dominate the growth of the cluster population.*

**Related to the overall picture and schematics of Fig. 8, authors remark that "conditions for different growth mechanisms depend on vapor properties and environmental conditions and thus they cannot be generalized to arbitrary compounds and conditions". However, as authors have performed simulations using two different scenarios for particle loss with different size dependencies, it would be interesting to know how sensitive this scheme is to the nature and strength of particle losses.**

Particle losses may affect the growth dynamics in different ways. For instance, if the magnitude of losses is increased or the size-dependence changed so that also larger particles are more easily scavenged, higher absolute vapor concentrations are needed for the particles to grow fast enough to reach activation sizes before being lost. Higher losses may also make coagulation among the clusters less significant. On the other hand, the effects on the whole distribution may be very non-linear if there are significant evaporation fluxes from the larger clusters towards the smaller sizes. However, although large changes in particle scavenging may change the growth dynamics, we can expect that the qualitative picture of Fig. 8 is not significantly affected by minor changes in the losses. We now note this in the manuscript (P6, L3):

*One should note that if cluster losses were significantly changed in our simulations, the growth dynamics of the cluster population could change. However, minor changes in the losses are not expected to affect our results on the qualitatively level.*

**Table 3 gives a good summary on different simulation sets, and it would help the reader if these would also be referred accordingly in Results and discussion.**

We followed the suggestion and added references to Table 3 in Results and discussion.

**Specific comments and technical corrections**

**Page 1, line 27: 'aerosol forcing' → 'aerosol radiative forcing'.**

We fixed this.

**Page 2, lines 1–12: The role of ions for the NPF process in addition to organic compounds and bases could be mentioned. Related to this, there is no reference corresponding to 'Kirkby et al., 2016' in the list of references.**

We added Kirkby et al. (2016) in the reference list and a following sentence discussing the role of ions in NPF (P2, L4):

*In addition, electric charge may enhance clustering when electrically neutral clusters are unstable or vapor concentrations are low (Lehtipalo et al., 2016; Kirkby et al., 2016).*

**Starting from page 2, there are several references to articles 'Kulmala, 2004' and 'Kulmala et al., 2004' in the manuscript. However, only the one corresponding to 'Kulmala, 2004' is given in the list of references, although it should be 'Kulmala et al., 2004'. The list of references has also other issues and should be revised by the authors.**

We fixed this and checked the list of references.

**Page 3, line 11: Heterogeneous, not homogeneous, nucleation of the organic vapour should be implied.**

It is true that Wang et al. (2013) focuses on heterogeneous nucleation. However, we are referring to nucleation more broadly here, regardless of the nucleation mechanism. Therefore, we decided to omit "homogeneous" from this sentence and simply write "*by nucleation*".

**In relation to Eq. (1), $a_{org}$ is used to denote the activity coefficient of the organic compound. As $a$ is commonly used for the activity, this seems somewhat misleading. I would recommend using $f_{org}$ for the activity coefficient, as $\gamma$ has been already reserved for other use. Also, the surface tension in the Kelvin term should refer to the droplet/cluster as whole, not to the organic compound.**

We changed $a_{org}$ to $f_{org}$ and removed "org" from the subscript of the surface tension term.

**Page 4, lines 17–18: It should be noted that if adsorption of vapour on the insoluble seed surface is taken into account, it is possible to have a maximum in the saturation ratio vs. cluster size curve [1].**

We modified the sentence to clarify that we are referring here to classical heterogeneous nucleation, where adsorption is not considered. The sentence now reads "*Note that this behavior is different than in classical heterogeneous nucleation,…*"

**Page 4, lines 18–30: Although mainly phrased in terms of water vapour, theoretical and simulation results can be found from the literature focusing on the nucleation/activation-transition [3, 4, 5], some of which might be relevant for discussion here.**

We added a citation to Reiss and Koper (1995) on P4, L16, where stable and unstable equilibrium are discussed.

**Figure 1: This is a very good figure illustrating the differences between simplified nano-Köhler theory and the real system behaviour. However, the meaning of double-headed thin arrow in the real system description is not clear, does it imply forward and backward crossing of the thermodynamic barrier?**

We apologize for the unclarity; the purpose of the arrow was to depict the coagulation of the two clusters. We have now modified the figure to better illustrate the coagulation process that can, indeed, lead to the crossing of the barrier.

**Table 1: Is there any reason, why the condensing organic vapour has to be water-soluble in the nano-Köhler theory?**

In the original nano-Köhler theory presented by Kulmala et al. (2004), the organic compound is assumed to be water-soluble to simplify the thermodynamic description of the system. In principle, the organic species could of course be insoluble, which would lead to different cluster/particle thermodynamics, that is, different evaporation rates. This does not change the qualitative picture regarding the role of the organic vapor in cluster growth, if the evaporation rate of the organic compound is still high for the small clusters, and decreases with cluster size.

**Page 6, line 8: General Dynamic Equation (not Dynamics). Also, although this is a matter of taste, Eq. (2), when given in molecular resolution, could be referred as an extended Smoluchowski coagulation equation, considering Marian Smoluchowski's seminal contribution to the theory.**

We changed "Dynamics" to "*Dynamic*". We also now mention that Eq. (2) is called Smoluchowski coagulation equation.

**Page 6, line 15: An original reference [2] for the ACDC model should be given.**

We added the reference suggested by the referee. It must be noted, though, that this reference describes only the very first version of the model, and most model features have been implemented in later versions. Therefore, we also refer also more recent work.

**Equation (3): Although containing the Kelvin term, this equation could be more properly referred as a condition of detailed balance than the Kelvin formula.**

This is correct. However, we prefer to refer to Eq. (3) as the Kelvin formula for two reasons: (1) In the atmospheric aerosol community, many readers are more familiar with this expression. (2) We would like to emphasize that we approximate the evaporation rates based on the classical Kelvin-Raoult description, instead of, for example, using Gibbs free energies of cluster formation obtained from quantum chemical calculations (which are unfortunately not available for large organic clusters).

**Page 7, line 16: Mass of 500 amu is given here for the LVOC$_{large}$, while in Table 2 and caption of Fig. A6, 600 amu are given. Which one is right?**

We apologize for the confusion; the mass of 600 amu is correct. We fixed the typo.

**Page 7, lines 26–29: Would there be other likely contributions, besides Raoult's law effect for the organic vapour, from the inclusion of water vapour into simulations?**

For the Kelvin-Raoult approximation and assumptions applied in this work, water would simply decrease the evaporation of the organic species. However, in reality water can also affect the surface

tension and density of the droplets (in the exponential factor of Eq. (3)), as well as the activity coefficient (Eq. (1)) when assuming non-ideal mixing. Furthermore, water molecules increase the cluster size, thus affecting the collision coefficients, and consequently also the evaporation coefficients through the detailed balance (Eq. (3)). This effect may however be minor compared to the effects on the thermodynamics. We added discussion about the additional effects of water in the manuscript (P7, L29):

*One should note, though, that in reality water can also affect the surface tension and density of the clusters as well as the activity coefficient when assuming non-ideal mixing. Furthermore, water molecules increase the cluster size, thus affecting the collision coefficients, and consequently also the evaporation coefficients through the detailed balance (Eq. (3); see also Henschel et al., 2016).*

**Table 3 and page 21, line 5: The unit for pressure (Pa) is missing.**

We added the missing units.

**Figures 2 and 5: Does the solid line indicate clusters with 1:1 stoichiometry? I could not find any explanation from the text.**

Yes, it does. We added an explanation for the line in the figure captions.

**Page 19, lines 13–15: When considering Eq. (3), this is right when considering a given compound. However, as in general higher molecular mass implies smaller equilibrium vapour pressure, this statement sounds odd. It should be noted that the ratio $m_{org}$ / $\rho_{org}$ in Eq. (3) refers to the (partial) molecular volume of the organic compound in the cluster, correlating strongly with the surface area of a (spherical) molecule at the surface. From this perspective, it might be better to rephrase the sentence pointing out the importance of molecular volume/exposed surface area instead of molecular mass on the equilibrium vapour pressure over a curved surface.**

We modified the sentence according to the referee's suggestion. The sentence now reads:

*The main reason for the difference is that the evaporation rate is higher for the compound with a higher mass, due to a larger molecular volume ($m_{org}/\rho_{org}$) and thus also a larger surface area of a molecule at the cluster surface (see Eq. 3).*

**Reply to Referee #2**

**"Exploring the potential of the nano-Köhler theory to describe the growth of atmospheric molecular clusters by organic vapors," by Kontkanen et al., explores the conditions in which nano-Kohler theory can be applied to the initial stages of new particle formation (NPF). The development of simple models that accurately represent NPF and the subsequent growth of nanometer-sized particles are needed in order to assess the importance of NPF in climate and air quality. This study can potentially address these needs, and therefore is quite appropriate for publication in ACP. I do, however, have one major concern that the authors should address before recommending publication. In addition, I will recommend a number of minor corrections.**

**This study has, as its main objective, the determination of whether nano-Kohler theory may be appropriate for representing NPF for range of compounds (H2SO4, LVOC, and ELVOC) and**

**concentrations (1E6 – 1E8) that are representative of ambient air in many locales. In order to test their implementation of nano-Kohler, the authors compared their results to those of a cluster kinetics model. Herein lies my concern. Since the authors use their comparisons between nano Kohler and cluster kinetics models as their metric for whether nano-Kohler is appropriate for describing atmospheric NPF, this paper should be more appropriately titled "Exploring the potential of the nano-Köhler theory to describe the growth of atmospheric molecular clusters by organic vapors as predicted by a cluster kinetics model." I assume that the authors wish the readers to interpret these results more generally, i.e., associate the information shown in Figure 8 (which, as an aside, is a wonderful graphic!) with actual atmospheric concentrations of H2SO4 and organics. But this is not what's being tested, nor have the authors placed effort into convincing the reader that the assumptions made in implementing the cluster kinetics mode actually result in an accurate description of atmospheric NPF.**

To clarify that our aim is to study the suitability of nano-Köhler theory to describe NPF based on cluster kinetics simulations, we changed the title of the manuscript to "*Exploring the potential of the nano-Köhler theory to describe the growth of atmospheric molecular clusters by organic vapors using cluster kinetics simulations*". However, we would like to point out that rather than focusing on the quantitative results obtained for the studied model systems, we wish to compare the predictions of the very simplified nano-Köhler description to those given by a full cluster kinetics model using the same input parameters for both models (as discussed below). The cluster population model includes processes that cannot be included in the nano-Köhler framework, e.g. the fact that even when the thermodynamic barrier prevents spontaneous condensation of the organic vapor, the organic compound may still be taken up by the clusters through nucleation (Fig. 1).

**I see two possible ways to address this issue, both of which could ideally be applied to this study. The first is to address my concern about the accuracy of cluster kinetics modeling for describing NPF under the range of conditions that are the foci of this study. Rather than assuming that the reader interprets the results of cluster kinetics modelling as "truth," the authors need to provide clear evidence of this fact. This includes the validity of the various assumptions used in that model, such as hard-sphere collisions, evaporation rates using Kelvin Theory, etc.**

We would like to clarify that we are not implying that the parameters that we use as input in our model are "truth". For instance, evaporation rates derived from Kelvin equation can significantly differ from the real evaporation rates. However, our aim in this study is not to use as chemically detailed input data as possible in the model simulations. Instead, we want to compare how well nano-Köhler theory, which describes the growth of cluster population in an extremely simplified way, compares with the cluster kinetics model, which gives a detailed and accurate description of the behaviour of cluster population, when the same input parameters are used. In other words, instead of addressing detailed chemistry, we focus on cluster population dynamics to explore how large effects these processes, which are omitted in nano-Köhler theory, may have on observed cluster growth.

To clarify this in the manuscript we modified the end of introduction (P3, L23) which now reads:

*In this study, our aim is to investigate the potential of nano-Köhler theory to describe the initial growth of atmospheric molecular clusters by organic vapors considering the complex dynamics of a cluster population. For this we use a molecular-resolution model, which allows us to explicitly simulate the time-evolution of a cluster population involving organic and inorganic species. First, we discuss similarities and differences between nano-Köhler theory and the traditional Köhler theory and compare their assumptions to real atmospheric molecular systems. Then, we apply cluster kinetics simulations to study the conditions under which nano-Köhler type behavior can be observed assuming representative molecular systems. Specifically, we investigate the effects of vapor properties, such as volatility and vapor concentrations, on the dynamics of the cluster population.*

*We also compare the results on cluster activation obtained from the detailed simulations to the predictions of the nano-Köhler theory. This way we can assess to what extent nano-Köhler theory, which describes the behavior of the cluster population in a very simplified manner, is able to capture the cluster growth.*

In addition, we added two sentences discussing the assumptions about evaporation rates (P6, L23):

*One should note that these evaporation rates can significantly differ from the real cluster evaporation rates in a system involving sulfuric acid and organic compounds. However, in this study our aim is not to use as complex evaporation rate data as possible but to compare nano-Köhler theory and cluster kinetics simulations with similar assumptions for evaporation rates.*

Finally, we now mention in the conclusions that the quantitative results on the conditions under which activation occurs depend on vapor and cluster properties (P21, L5):

*However, it must be kept in mind that the quantitative results depend on the exact vapor and cluster properties.*

We would also like to point out that in the end of the conclusions, we state that improved understanding of cluster thermodynamics, including composition and size-dependent evaporation rates, is needed. Thus, we trust that the uncertainties in our model parameters are clear to the reader.

**My other recommendation is to use experimental data to compare to the results of nano-Kohler. Prior studies have explored cluster growth rates as a function of measured H2SO4 concentrations (e.g., "Size and time-resolved growth rate measurements of 1 to 5 nm freshly formed atmospheric nuclei," Kuang et al., ACP, 2012), so it would seem a simple task to take measured growth rates and [H2SO4] and explore the predicted growth rate from nano-Kohler using realistic assumptions for [LVOC] and [ELVOC]. My first recommendation, I feel, is necessary for this paper. My second recommendation would allow readers to have a lot more confidence that the data shown in Figure 8 is truly representative of the real atmosphere.**

We feel that there is a misunderstanding regarding the purpose and methods of our work. In this study, we address the fact that the simplified nano-Köhler theory might not be a suitable approach to interpret observed ~sub-5 nm nanoparticle formation, because the population of these small clusters is affected by various dynamic processes not included in nano-Köhler theory (see Sect. 2) The cluster population simulation data can be considered as synthetic "measurement" data, against which nano-Köhler theory is validated using the same cluster properties in the simulations and in the nano-Köhler predictions. The fact that the cluster population does not always behave according to the nano-Köhler predictions demonstrates that nano-Köhler theory is not necessarily capable of capturing the details of cluster growth, even if all parameters used in nano-Köhler calculations were exactly correct. However, we now further emphasize (e.g. in Conclusions) that the quantitative results depend on the parameters related to e.g. cluster stability (see the reply to the previous point above), as the Reviewer rightfully points out.

In this work, we also demonstrate and discuss the problems related to interpreting apparent ~sub-5 nm growth rates, which are deduced from the cluster simulation data according to the standard experimental approach (the appearance time method) (e.g. Sect. 4.2, 4.3.1 and 4.3.2). The behaviour of the apparent growth rate of the population may be linked to an activation process, but may also be due to other population dynamics processes. Therefore, interpreting experimentally determined growth rates through nano-Köhler theory, or fitting a nano-Köhler model to the growth rates, is likely to involve considerable uncertainties – or result in getting the observed growth right for wrong reasons.

In general, we feel that comparing our modelling results to experimental data is outside the scope of this study. Quantitative comparison would be challenging due to significant uncertainties in the large number of model parameters and in the exact properties and concentrations of organic compounds detected in field and laboratory experiments. However, in the future we are planning to perform cluster population simulations applying more detailed cluster properties for specific chemical systems, which we aim to compare with experimental data.

**The following are minor suggestions, where each comment is preceded by the page**

**and line number.**

**P1, L29: Shouldn't the word "including" be replaced by "specifically"? Including suggests**

**that the phrase that follows is a process that differs from NPF, but in my view it specifically defines NPF. In general I would recommend to the authors that they do a better job of defining, very early in the manuscript, what is meant by NPF. In this paper, the focus is on the formation of the cluster and the growth up to a few nanometers in diameter. One gets that point later in the paper, but I feel it could be made more clear from the start (this includes the abstract).**

We changed "including" to "*specifically*" and added the following sentence in the beginning of the abstract: *Atmospheric new particle formation (NPF) occurs by the formation of nanometer-sized molecular clusters and their subsequent growth to larger particles.*

In addition, to clarify that we focus on the very first steps of NPF, we modified two sentences in the abstract (P1, L18) and in the introduction (P3, L17) by adding "*initial*" in front of the "the growth of atmospheric molecular clusters".

**P3, L21: "to study in what kind" is awkward phraseology. I suggest "to study the conditions**

**under which"**

We rephrased the sentence following the referee's suggestion.

**P3, L26: "The nano-Kohler" does not require the article "The" . . . this is a common error**

**throughout the manuscript.**

We fixed this.

**P4, L10: The term "seed cluster" is used here but it really hasn't been introduced. What**

**is a seed cluster and why is it required?**

In nano-Köhler theory seed clusters refer to the initial molecular clusters which can become activated to growth by organic vapors. To clarify this, we added "*initial*" in front of the "seed cluster" on P4, L10. In the absence of seed clusters or other clustering compounds, the initial cluster formation can occur only by organic vapors. If pure organic clusters are highly unstable, clustering is very weak and particle formation does not occur, except possibly at high organic vapor concentrations. As discussed in Sect. 2, in real systems of atmospheric molecular clusters there is no specific non-evaporating seed or condensing vapor but there is a distribution of inorganic and organic vapor molecules and clusters which all can collide and evaporate.

**P6, L27: This stated loss rate due to dilution is unique to the CLOUD experiments, as it depends on the flow rates into and out of the chamber, that authors should state this fact.**

We modified the sentence to clarify this issue. The beginning of the sentence now reads "*In most simulations the external loss coefficient $L_i$ was set to correspond to losses in the CLOUD (Cosmics Leaving OUtdoors Droplets) chamber,..*"

**P7, L26: Wouldn't adding water content to the model also increase uptake of some compounds such as H2SO4 and other hygroscopic organics, due to increased surface area?**

Yes, this would happen if the compounds are assumed to be hygroscopic. In reality water can also affect the surface tension, density and activity coefficient of the clusters. Water molecules can also increase cluster sizes and thus affect both the collision and the evaporation rates. See also the answer to the comment by Referee #1 related to the effects of water.